# Essential Oils as a Potential Neuroprotective Remedy for Age-Related Neurodegenerative Diseases: A Review

**DOI:** 10.3390/molecules26041107

**Published:** 2021-02-19

**Authors:** Aswir Abd Rashed, Ahmad Zuhairi Abd Rahman, Devi Nair Gunasegavan Rathi

**Affiliations:** 1Nutrition, Metabolism and Cardiovascular Research Centre, Institute for Medical Research, National Institutes of Health, Ministry of Health Malaysia, No.1, Jalan Setia Murni U13/52, Seksyen U13 Setia Alam, Shah Alam 40170, Malaysia; rathidevinair@moh.gov.my; 2Cancer Research Centre, Institute for Medical Research, National Institutes of Health, Ministry of Health Malaysia, No.1, Jalan Setia Murni U13/52, Seksyen U13 Setia Alam, Shah Alam 40170, Malaysia; zuhairirahman@gmail.com

**Keywords:** essential oils, neurodegenerative, Alzheimer’s disease, Huntington’s disease, Parkinson’s disease, amyotrophic lateral sclerosis, in vitro, in vivo

## Abstract

Despite the improvements in life expectancy, neurodegenerative conditions have arguably become the most dreaded maladies of older people. The neuroprotective and anti-ageing potentials of essential oils (EOs) are widely evaluated around the globe. The objective of this review is to analyse the effectiveness of EOs as neuroprotective remedies among the four common age-related neurodegenerative diseases. The literature was extracted from three databases (PubMed, Web of Science and Google Scholar) between the years of 2010 to 2020 using the medical subject heading (MeSH) terms “essential oil”, crossed with “Alzheimer’s disease (AD)”, “Huntington’s disease (HD)”, “Parkinson’s disease (PD)” or “amyotrophic lateral sclerosis (ALS)”. Eighty three percent (83%) of the studies were focused on AD, while another 12% focused on PD. No classifiable study was recorded on HD or ALS. EO from *Salvia officinalis* has been recorded as one of the most effective acetylcholinesterase and butyrylcholinesterase inhibitors. However, only *Cinnamomum* sp. has been assessed for its effectiveness in both AD and PD. Our review provided useful evidence on EOs as potential neuroprotective remedies for age-related neurodegenerative diseases.

## 1. Introduction

Aromatic plants consist of a wide and diverse array of organic compounds with significant ecological and physiological functions. One of the most vital components synthesised by aromatic plants are essential oils (EOs), along with its secondary metabolites and phenolic compounds [1]. EOs can be extracted and obtained from various parts of plants, such as the flower, bark, leaf, root, or peel [2,3,4]. Generally, monoterpenes and sesquiterpenes are the main constituents of EOs. Phenolic compounds are generated via biochemical synthesis and consist of a chemically heterogeneous group. Phenolic acids, simple phenols, coumarins, flavonoids, stilbenes, lignans, lignins, as well as hydrolysable and condensed tannins are among the well-established phenolic compounds [5,6]. EOs are volatile and they may play a role in cognitive improvement through olfactory pathways [7]. EOs are well-known for various benefits that include its antiviral, antibacterial, antifungal, memory enhancement, medicinal remedy, food preservation, cosmetic preservative, aromatherapy, and many other applications. For example, EO sourced from *Salvia* sp., which is one of the most common medicinal plant species, was reported for its notable remedy in cough, bronchitis, herpes, thrush wounds, as well as in impaired concentration. The EO of this species is also applied in the food industry and cosmetic industry, for ranges of perfume products [8]. 

The brain’s central nervous system (CNS) is composed of diverse neurons responsible for the organisation of neuronal and non-neuronal cells, as well as handling various motor, sensory, regulatory, behavioural, and cognitive functions. The neuronal cells are diverse in their morphology and function, suggesting that each neuronal type may indicate its own genomic profile despite having identical genetic codes. Within the CNS, specific regions were noticed to exhibit different vulnerabilities to ageing and various age-related neurodegenerative diseases [9].

Neurodegenerative disorders are often characterised by strong evidence of oxidative stress in their pathogenesis, as a consequence of unregulated synthesis of reactive oxygen species (ROS) [10]. Disparity observed in pro-oxidant and antioxidant cellular mechanisms inter-related with mitochondrial dysfunction, lipid peroxidation, neuroinflammatory processes, and endogenous dopamine metabolism are among the contributing factors to deregulation [10,11]. Many researchers have searched for molecules that activate blocking pathways or minimise the effects of ROS [12,13]. In an effort to overcome the limitations of current therapeutics available for neurodegenerative disorders, substantial research is being undertaken to explore and identify the availability of other possible natural drugs that are equally effective and without any side effects. As such, natural components composed of various polyphenolic phytochemicals have gained notable insight for this purpose [14].

Neurodegenerative disorders are currently incurable, and the available therapies only control symptoms or prolong the disease’s growth. EOs have been proposed as an underlying preventive and treatment strategies for anti-ageing and neurodegenerative disorders [15]. Many studies have reported the potential of various EOs and their components to exhibit neuroprotective effects [16,17]. *Cinnamomum* sp. [18,19], *Salvia* sp. [20,21], *Polygonum* sp. [22], *Lavandula* sp. [23,24], *Citrus* sp. [25], *Artemisia* sp. [26,27], and *Zingiber* sp. [28] are among the most widely explored species for evaluating the effectiveness of EOs and its respective components in age-related neurodegenerative disorders. The four most commonly studied age-related neurodegenerative diseases are Alzheimer’s disease (AD), Parkinson’s disease (PD), Huntington’s disease (HD) and amyotrophic lateral sclerosis (ALS).

AD is the most common cause of dementia in the elderly and is classified as a slow yet progressive neurodegenerative disorder. The highest prevalence rates are reported in North America and Western Europe followed by Latin America, China, and the Western Pacific. In general, it is highlighted that large number of AD cases are noticed among elderly people aged over 75, however early-onset of AD can also develop as early as 30 up to 60 years [29,30]. The direct cost entailed to AD diagnosis covers medical treatment or social services where a caregiver is needed, while a patient’s or family members’ income loss is referred to as indirect cost [31].

There are several AD profiles, which include deficits in episodic memory, language, semantic knowledge, visuospatial abilities, executive functions in terms of planning and organisation as well as apraxia [32]. Apart from neuronal loss, amyloid plaques and neurofibrillary tangles are inter-related to the presence of reactive astrocytes and activated microglial cells [33,34,35]. Aβ is the most widely studied component of AD pathogenesis, where it can induce neuronal toxicity and activate microglia leading to the indirect damage of neurons [36]. Proteolytic cleavage from the type I cell-surface protein amyloid precursor protein (APP) was known to yield several forms of Aβ [37,38]. The pathogenic hypotheses for synaptic and neuronal toxicity in Alzheimer’s disease is shown in Figure 1.

A significant change that correlates to the age-related loss of substantia nigra (SN) dopaminergic cells is the loss of neostriatum dopaminergic innervation. Compelling studies have shown that the involvement of monoamine oxidase (MAO) in AD and neurodegenerative diseases is an important factor in many major pathophysiological pathways [40,41]. MAO-B has been suggested as a biomarker, and its activated form leads to cognitive dysfunction, kills cholinergic neurons, induces cholinergic disorders, and contributes to the development of amyloid plaques. Studies in molecular biology have demonstrated the critical role of Aβ generation through the modulation of the processing of APP by MAO [42,43,44,45]. The mechanism of Aβ generation through modulation of APP processing by activated MAO is shown in Figure 2.

PD is the second most prevalent condition after AD, and also develops slowly over time [47]. PD can be identified and clinically characterised via motor impairment, which includes bradykinesia, rigidity, resting tremor and postural instability [48]. PD cases can be divided into sporadic (sPD) and familial (fPD), the latter of which represents approximately 20–25% of all PD cases. A common hallmark of sPD and fPD is the presence of intracellular inclusions, termed Lewy bodies [49,50].

α-Synuclein (α-Syn) has been identified as a major component of Lewy bodies in sporadic and familial cases, and is believed to be the central player in PD aetiology [51]. It is worthwhile to note that research conducted on PD has mainly focused on protein aggregation, neurotoxicity, increased oxidative stress, and mitochondrial dysfunction, as well as defects in the protein degradation machinery [52].

Apart from the role caused by α-Syn, the presence of neurotoxins, in particular 6-hydroxydopamine (6-OHDA) and 1-methyl-4-phenylpyridinium (MPP+), are widely accepted to induce neurotoxicity in PD patients. Both neurotoxins are thought to induce dopaminergic toxicity by intra- and extracellular oxidation, hydrogen peroxide formation, and direct inhibition of the mitochondrial respiratory chain [53].

The next common neurodegenerative disease is HD; caused by the recurrent development of cytosine–adenine–guanine (CAG) in the huntingtin (HTT) gene and involves a network of complex pathogenic mechanisms. HD is a profoundly penetrating, autosomal dominant, progressive neurodegenerative movement and neurobehavioural disorder associated with a variety of motor signs, psychological symptoms and cognitive dysfunction that progress with dementia. Knowledge of HD’s causal mutation allows the detection of an ever-expanding number of HD phenotypes and phenocopies. The mean starting age is around 40 years, with a recorded range of 2 to 79+ years [54,55,56,57]. 

Progressive physical impairment of HD could be contributed to by various movement aspects such as hyperkinetic movements (dystonia, myoclonus, tics) and other motor manifestations (bradykinesia, incoordination, oculomotor function changes, gait impairment) along with chorea as the most distinct involuntary motion. As the disease progresses over time, dystonia becomes more prevalent and replaces chorea.

On the other hand, ALS is categorised as a heterogeneous neurodegenerative condition clinically, genetically, and pathologically [58,59,60]. The degeneration of cortical motor neurons and anterior horn cells of the spinal cord is characterised by ALS, also known as Charcot’s or Lou Gehrig’s disease. This contributes, usually within 3–5 years of diagnosis, to muscle atrophy, loss of muscle function, and death resulting from respiratory failure. The diverse clinical variability in ALS is believed to be due to differences in upper motor neuron (UMN) and lower motor neuron (LMN) involvement, extra-motor symptoms, onset age, survival, and progression rates. Heterogeneity of the disease prevents biomarker production which hinders the accurate evaluation of candidate drugs in clinical trials [59,60,61]. Various studies have shown that oxidative stress plays a major role in this disease’s pathogenesis, identified as an unusual family type that often exhibits superoxide dismutase 1 (SOD1) gene mutations [62,63].

EOs are now commonly used and their advantages in all facets of life are investigated. The attractiveness of EO potential and possible mechanism of actions are ventured on a continual basis around the globe. In this review, we focus on and analyse the effectiveness of EOs as neuroprotective remedies among the four selected age-related neurodegenerative disorders mentioned above. We strongly believe that this review will be beneficial to many researchers and academicians that have great interest in EOs and their extensive applications. 

## 2. Materials and Methods

### Search Strategy

Original articles were searched in three databases (PubMed, Web of Science and Google Scholar) from the year 2010 to 2020, using the medical subject heading (MeSH) terms “essential oils”, crossed with the term “Alzheimer’s disease”, “Parkinson’s disease”, “Huntington’s disease” or “amyotrophic lateral sclerosis”. Publications with available abstracts were reviewed and limited to studies published in the English and Malay languages. Papers on human and animal studies, clinical trials, and related to plant-based neurodegenerative medication were included. However, review articles and letters to the editor were excluded. Duplicate articles were eliminated. 

## 3. Results

All the related articles were printed out for further evidence-based assessment to explore the effectiveness of EOs as a neuroprotective remedy for age-related neurodegenerative diseases. After conducting the comprehensive literature review, the articles were selected and divided into several types of neurodegenerative diseases (Table 1). A total of 103 articles were included in this review. Eighty-six articles on AD were found, which consisted of 53 articles on in vitro studies, 20 articles on in vivo studies, 11 articles on the combination of in vitro and in vivo studies, and 2 articles on a combination of in vitro and ex vivo studies. Thirteen articles on PD were found, which consisted of four articles on in vitro studies, six articles on in vivo studies, one article on the combination of in vitro and in vivo studies, and two articles on a combination of in vivo and ex vivo studies. Four articles were categorised as a combination of diseases because several neurodegenerative diseases were mentioned in the articles at once. Unfortunately, no classifiable study was recorded on HD and ALS. Besides the standard biochemical assay, some studies also reported on the chemical composition of the selected EOs. 

## 4. Discussion

EOs contain the essence of different scents and the properties from their originating plants. These volatile oils display various biological activities [154]. They are mainly used in the beverage, fruit, cosmetic, and fragrance industries [155]. EOs derived from steam distillation process are mainly used in pharmacological activities and food products, while the extracts from lipophilic solvents are utilised in the fragrance industry [156]. Several EOs have been well-known for their usage in fragrances and flavours for hundreds of years. EO usage in the fragrance industry is mainly due to by their attractive odour. The extensive benefits offered by EOs signify the continuous demand that is seen to be increasing steadily.

### 4.1. The Source of EOs

As mentioned before, EOs can be extracted and obtained from various parts of plants. Clove’s EO derived from the *Syzygium aromaticum* tree’s aromatic flower buds with origin from Maluku, Indonesia contains the powerful scent used in spiced foods [154,155]. *Eucalyptus globulus* oil is mint-like, with properties such as a decongestant, pain relievers, antimicrobial agent, immunostimulant, flu and cold/cough treatment, as well as for mental clarity in aromatherapy [157,158]. One of the most influential EO is from *Lavandula angustifolia,* which is also known as English lavender. Lavender oil possesses strong antioxidant, anti-inflammatory, antibacterial, and antimicrobial properties, and can be used to treat various skin diseases (e.g., eczema, ringworm, acne), improve digestive system, minimise sore muscle swelling, and additionally, alleviate pain [157,158]. *Citrus limon* EOs are used as antimicrobial and antifungal agents, pain relievers, aids in weight loss, and to reduce extreme nausea as well as for usage as soaps, hair shampoo, furniture polishes, and fresheners [157,158]. Oregano (*Origanum vulgare*) EOs are often used for skin care, menstrual problems, stomach problems, and to control flu and cold infections [157,158]. Rosemary’s EO originates from the *Rosmarinus officinalis* evergreen shrub with characteristics of a crisp woody, herbal and balsamic odour, similar to camphor. The usage of rosemary oil ranges from various treatments of skin care, dandruff, and scalp health, as well as to cold prevention and boosting the immune system [157,158]. EO from *Mentha piperita* is called peppermint oil, which is mainly used in the prevention of flu and colds, reduces headache symptoms, and also in relieving muscle and joint pains [157,158].

Apart from the general usage of EOs, their extensive benefits have also been noticed and reported in relevance to age-related neurodegenerative disorders. Based on available studies, EOs have been proposed as an effective preventive and treatment approach for anti-ageing and neurodegenerative disorders. Therefore, we attempted to describe and highlight the various EOs, and the effectiveness of their components with respect to the four common neurodegenerative diseases (AD, PD, HD, and ALS), as mentioned above. The different parameters that are commonly used for the evaluation of each disease are explained, accordingly. 

Based on Table 1, a total of sixty-nine types of EOs from different genera of plants were evaluated for their effectiveness against neurodegenerative diseases among studies conducted between 2010 and 2020. In reference to all the compiled literature, we observed that the IC_50_ results were presented in several formats, in particular to the units of measurement. Among the units that were reported included mg gallic acid equivalents/g [64,76], percentage [65,74], mg/mL or mg/L [26,67,68,73,75], μm [69], and µg/mL [18,70,71,72]. This diversity, however, was considered troublesome because direct comparison among studies with different measurement units is not possible without conversion.

### 4.2. Major Component of EOs

The chemical structures of several major components commonly found in EOs that have been reported to have anti-neurodegenerative properties are presented in Figure 3. Based on our review, 1,8-cineole has been identified as one of the major components found from various types of EOs. The compound 1,8-cineole is a saturated monoterpene that can originate from several plant species (e.g., *Eucalyptus*, *Rosmarinus*, and *Salvia*), with *Eucalyptus* leaves recognised as the key source [159]. Sometimes called eucalyptol due to its natural source, 1,8-cineole should not be confused with eucalyptus oil, a combination of many other components [160]. Due to its excellent aroma and taste, 1,8-cineole is mostly used in fruit, fragrances, and cosmetics. Furthermore, pure monoterpene 1,8-cineole is used as an alternative sinusitis remedy for respiratory tract infections, such as common cold or bronchitis [161]. It was indicated as one of the most potent free radical scavengers that may influence anticholinesterase activity based on a study reported by El Euch and colleagues [88] The antioxidant activity was measured using the free radical 1,1-diphenyl-2-picrylhydrazyl (DPPH) test. Essential oil concentration providing 50% inhibition (IC_50_) of the initial DPPH concentration was calculated using the linear relationship between the compound concentration and the percentage of DPPH inhibition. Ascorbic acid was used as a standard. In the study by Abuhamdah et al. [66], EO extracted from the leaves of *Aloysia citrodora Palau* showed neuroprotective activity and a higher presence of 1,8-cineole was reported (23.66%). 

In another study performed by Cutillas and his team [97] on EO from *Thymus mastichina L*., it was asserted that among all four compounds (α-pinene, β-pinene, limonene and 1,8-cineole), 1,8-cineole was the best AChE inhibitor with an IC_50_ of 35.2 ± 1.5 μg/mL. They tested the antioxidant activity using five different methods such as the oxygen radical absorbance capacity (ORAC) assay, the 2,2′-azino-bis (3-ethylbenzothiazoline-6-sulphonic acid) (ABTS) antioxidant method, the 2,2-diphenyl-1-picrylhydrazyl (DPPH) method, the thiobarbituric acid reactive substances (TBARS) method, and the chelating power (ChP) method. All the methods recorded that 1,8-cineole was one of the compounds highest in antioxidant capacity. This finding was similar to several other types of research, which suggested the efficacy of 1,8-cineole as the dominant anticholinesterase agent from EOs of different sources [8,21,25,152].

### 4.3. Cholinesterase Activity in EOs

The most common criteria used in the determination of AD is related to anti-cholinesterase activity. Cholinesterases (ChEs) are specialised carboxylic ester hydrolases that catalyse the hydrolysis of choline esters. Two types of ChE activity have been identified in mammalian blood and tissues, which are distinguished according to their substrate specificity and sensitivity to selective inhibitors. The first is acetylcholinesterase (AChE), which is systematically known as acetylcholine acetylhydrolase [172]. The second is butyrylcholinesterase (BChE), which is systematically referred to as acetylcholine acyl hydrolase [173,174,175,176]. The preferred substrate for AChE is acetylcholine (ACh), while butyrylcholine (BCh) and propionylcholine (PCh) are ideal for BChE [175,176,177]. AChE activity is known to be inhibited by several compounds, with toxins and drugs as the major inhibitors [178]. AChE activity is used in verifying treatment effects, especially in AD [177].

Both AChE and BChE possess active sites at the bottom of 20 Å-deep gorges with 50% identical amino acid sequence, whereas the gorge entrance locates the peripheral site [179]. The active site for both enzymes comprises a catalytic triad, acyl-binding pocket, and choline binding site [180]. A total of 14 aromatic amino acids are found in the active site of AChE, whereas six of these are substituted by aliphatic amino acids for BChE [181]. Binding and hydrolysis processes of bulky ligands are restricted in AChE due to presence of phenylalanine residues in the acyl binding pocket. In contrast, these residues are substituted with two flexible amino acids that are selective for BChE and allow the binding of bulkier ligands [182]. The different mechanisms involved in relevance to the active gorge site specific for each enzymes have been investigated via molecular modelling, structure-based virtual screening, or even crystallographic studies [181,182,183,184].

Generally, traditional Ellman assay is used with some modifications, applied for the determination of anti-cholinesterase activities [185,186]. This technique is a simple, accurate, and rapid method of measuring ChE activity, that is based on the reaction between thiocholine with the sulfhydryl group of a chromogen such as 5,5′-dithiobis-(2-nitrobenzoic acid) (DTNB or Ellman’s reagent). The shift of electrons to sulphur atoms yields a yellow substance called 5-thio-2-nitrobenzoic acid (TNB), which is measured by monitoring absorbance at 410 nm [187,188,189,190]. DTNB is a water-soluble compound and is useful for its fast reaction with thiocholine and minor side effects at neutral pH [185,187,188,189,190,191]. This technique, however, is also subject to certain limitations; it is restricted for testing antidots against organophosphorus AChE inhibitors or for measuring AChE activity in samples of such treated individuals [190]. In addition to the Ellman assay, another method that can also be used for measuring ChE activities is the electrometric method of Michel [192]. This technique is applied based on pH changes that arise from H^+^ synthesis via cholinester hydrolysis [175,176,193].

### 4.4. Extracellular Plaque Deposits

Extracellular plaque deposits of the Aβ-peptide and flame-shaped neurofibrillary tangles of the microtubule-binding protein tau are the two hallmark pathologies required for AD patients. Familial early-onset forms of AD are associated with mutations either in the precursor protein for Aβ (APP) or in presenilin-1 (PS1) or presenilin-2 (PS2). Peptide generation pathways synthesise γ-secretase with either PS1 or PS2 as the catalytic subunit. APP is sequentially cleaved, where β-secretase first cleaves APP to release a large, secreted derivative, sAPPβ, followed by γ-secretase that cleaves a fragment of 99 amino acids (CTFβ) to generate Aβ. The process of γ-secretase cleaving can be inaccurate, leading to C-terminal heterogeneity of the resulting peptide population that generates numerous Aβ species, with Aβ1–-40 of the highest abundance followed by Aβ42. The slightly longer forms of Aβ, particularly Aβ1–42, are the principal species deposited in the brain that are more hydrophobic and fibrillogenic [193]. In view of their vital role in Aβ synthesis, both β- and γ-secretase are considered as key components in anti-AD pharmaceuticals developments [193,194]. Normal pathology tests refer to the density in the affected brain regions of neuritic amyloid plaques and neurofibrillary tangles of tau protein. AD diagnosis involves the presence of large neuritic plaque portions, consisting of highly insoluble Aβ in the brain parenchyma. There are also deposits of tau protein, although they occur among less common neurodegenerative disorders, especially in the absence of neuritic plaques. There are some distinctive morphological features of the neurofibrillary tangles in the various diseases, and may exhibit a distinct composition of tau isoforms that vary from AD [195].

It is not only humans which have amyloid beta; non-human primates (NHPs) have the same Aβ sequences as humans, an almost identical APP sequence, and they overlap with related human biochemical pathways in many aspects, however surprisingly with ageing, they develop relatively few AD-like neuropathologies. Aged canines also develop severe amyloid deposition; canines tend to demonstrate extensive amyloid deposition from about ten years of age, unlike in aged NHPs, where it could take several decades [196]. Amyloid deposition in canines is also interrelated with age-related cognitive dysfunction [197], although little neuronal loss is detected. Due to a poor understanding of AD and the human brain complexity, it has been deduced that there is no natural animal model of the disease [198]. For the past 25 years, pharmacological and genetic AD-models, as well as various animal species (primates, dogs, rodents, etc.), have been used in AD research activities [199,200]. The resurgence of interest in rats as the appropriate animal model of AD led to the usage of various types of rat models. As of current practice, transgenic mice have been extensively used in studies on AD. In any selected models, all need the introduction of some combined familial AD mutation into APP or PS1, or even in both [200].

### 4.5. Current EOs on AD

*Salvia* is the largest genus of plants in the family Lamiaceae, with the number of species estimated to range from 700 to nearly 1000. Fifteen species of *Salvia* (namely, *officinalis* L., *chionantha*, *chrysophylla Staph*, *urmiensis*, *nemorosa* L., *syriaca*, *ballsiana*, *cyanescens*, *divaricate*, *hydrangea*, *kronenburgii*, *macrochlamys*, *nydeggeri*, *pachystachys*, *pseudeuphratica*, and *rusellii*) were studied for cholinesterase inhibition assay, and happen to be the most widely studied source of EOs. Most of the researchers found that *Salvia* spp. were weak AChE and BChE inhibitors, except for EO from *S. pseudeuphratica*, which demonstrated the highest inhibitory activity against AChE, at IC_50_ = 26.00 ± 2.00 μg/mL compared to *S. cyanescens* and *S. pachystachys*. In contrast, BChE activity did not show 50% inhibition even at the highest concentration, where IC_50_ was reported to be above 80 μg/mL for *S. pseudeuphratica* and *S. hydrangea* [96]. *Salvia officinalis* was studied twice, in 2017 and 2019. Based on GC–MS and GC–FID analysis, the main components of *S. officinalis* were α-thujone, camphor, 1,8-cineole, and β-thujone. *Salvia* has also been used in ex vivo-based research, using the isolated guinea pig ileum method where the major molecule, rosmarinic acid, showed significant contraction responses on an isolated guinea pig ileum. Docking results of rosmarinic acid also showed a high affinity to the selected target, AChE [136]. The author suggested the potential of rosmarinic acid to become a novel therapeutic candidate for the treatment of AD.

Other than *Salvia* spp., EOs from *Lavandula* spp. have also been studied for the treatment of AD. *L. luisieri* has been found to comprise high contents of oxygen-containing monoterpenes, mainly necrodane derivatives, which are absent from any other oil. This oil was tested on the endogenous beta-site APP-cleaving enzyme 1 (BACE-1) in cultured cells, being responsible for a reduction in Aβ production, with no significant toxicity. Although the study was conducted in vitro, the low molecular weight and high hydrophobicity of terpenoids are properties that provide a good chance for them to cross cellular membranes and the blood–brain barrier, an essential attribute for BACE-1 inhibition in vivo [78]. However, EO from *L. angustifolia* did not give a required finding because it enhanced the Aβ aggregation based on the thioflavin T method; this effect was further confirmed by atomic force microscope (AFM) imaging. EO of *L. angustifolia* was also showed to counteract the increase in intracellular reactive oxygen species production and the activation of the pro-apoptotic enzyme caspase-3 induced by Aβ1–42 oligomers [23,24]. Meanwhile, EO from *L. pubescens* exhibited strong anti-AChE and anti-BChE effects at IC_50_ of 0.9 μL/mL and 6.82 μL/mL, respectively. Carvacrol (CAR, 2-methyl-5-isopropylphenol) was also found to be higher in *L. pubescens*. Carvacrol was found to be abundant among EOs of the Lamiaceae family, and is known for various benefits including antibacterial, antifungal, antioxidant, antinociceptive, anti-inflammatory, anti-apoptosis, and anti-cancer activities [201]. Several studies on EOs have reported that carvacrol exerts some actions on the neuronal system, including AChE inhibition [104,202], anxiolytic [203], and antidepressant [204] properties. In addition, carvacrol has the ability to modulate central neurotransmitter pathways, such as dopaminergic, serotonergic and γ-aminobutyric acid (GABA)-ergic systems [201].

Only two types of cell lines, SH-SY5Y and PC-12, were reported to have been used in AD research. The in vitro toxic effects of amyloid peptides are usually examined using the human neuroblastoma-derived SH-SY5Y cell line, because differentiated neuron-like SH-SY5Y cells are extra-sensitive to amyloid peptides compared to non-differentiated cells, because the latter lack long neurites [205]. *Z*-ligustilide (Z-LIG) EOs effectively protect against fibrillar aggregates of Aβ25–35- and Aβ1–42-induced toxicity in SH-SY5Y and differentiated PC12 cells, possibly through the concurrent activation of the PI3-K/Akt pathway and inhibition of the p38 pathway [105]. Aβ25–35 represents a neurotoxic fragment of Aβ1–40 or Aβ1–42, and retains the toxicity of the full-length peptide [206]. Aβ25–35 is often selected as a model for full-length peptides because it retains both its physical and biological properties [207]. In general, declining levels of PI3K subunits as well as blunted Akt kinase phosphorylation have been observed in the AD brain, which is characterised by Aβ and tau pathologies [208].

There was also a study on the potential therapeutic effect of hybrid EO from Kushui roses. Kushui rose (*R. setate × R. rugosa*) refers to a natural hybrid of cog rose and traditional Chinese rose that has been cultivated for more than 200 years [209]. In this study, transgenic worm strains purchased from the Caenorhabditis Genetics Center (CGC) were used instead of rat or mice models. They found that rose EO (REO) significantly inhibited AD-like symptoms of worm paralysis and hypersensitivity to exogenous serotonin (5-HT) in a dose-dependent manner. Although the GC–MS analysis revealed the presence of 40 components, the major components, β-citronellol and geraniol, were found to act less effectively than the oil itself. Intriguingly, REO significantly suppressed Aβ deposits and reduced the Aβ oligomers to alleviate the toxicity induced by Aβ overexpression [209].

Su He Xiang Wan (SHXW) has also been studied for its neurodegenerative remedy potential. SHXW is a distinct EO, and is a patent medicine comprising borneol, styrax resin, musk, aquilaria, frankincense, piper, benzoin, saussurea, cyperus, sandalwood, clove, terminallia, aristolachia fruit, rhino horn, and cinnabar. This ancient prescription was recorded in the *He Ji Ju Fang* of the Song Dynasty [210]. For this plant, the researchers evaluated the effects of a modified SHXW (KSOP1009 formulation) intake on the AD-like phenotypes of Drosophila AD models, which express human Aβ1–42 in their developing eyes or neurons. They found that Aβ1–42-induced eye degeneration, apoptosis, and locomotive dysfunctions were strongly suppressed. However, Aβ1–42 fibril deposits in the Aβ1–42 overexpressing model were not affected by treatment with KSOP1009 extract. Conversely, KSOP1009 extract intake significantly suppressed the constitutive active form of hemipterous, a c-Jun N-terminal kinase (JNK) activator, while it induced eye degeneration and JNK activation. In Drosophila, flies with mutations that augment JNK signalling accumulate less oxidative damage and live dramatically longer than wild-type flies [211,212].

*Cinnamomum zeylanicum* consisting of (*E*)-cinnamaldehyde (CAL) (81.39%) and (*E*)-cinnamyl acetate (CAS) (4.20%) as the main compounds showed over 78.0% inhibitory activity in cholinesterase. In MAO-A and MAO-B inhibition assays, EOs from *C. zeylanicum* (96.44%, 95.96%) and CAL (96.32%, 96.29%) demonstrated comparable activity to rasagiline (97.42%, 97.38%, respectively). Research by Murata and co-workers [69] found that kaur-16-ene, nezukol, and ferruginol isolated from plants had anti-AChE (IC_50_) activity at 640, 300 and 95 μm, respectively. Even though ferruginol activity has already been highlighted before, by Gulacti et al. [213], this study documented the activities of kaur-16-ene and nezukol for the first time.

Meanwhile, *Citrus limon* has been found to significantly lower AChE brain depression in APP/PS1 and wild-type C57BL/6L (WT) mice. PSD95/synaptophysin, the synaptic density index, was substantially improved in histopathological shifts [109]. Based on the previous analysis by other researchers, nobiletin 3′,4′,5,6,7,8-hexamethoxyflavone was found to be the major component of polymethoxylated flavones in citrus peels, such as *C. depressa*, *C. reticulata*, *C. sinensis*, and *C. limon* [214,215]. Thus, nobiletin may potentially be the compound that substantially alters the development of these diseases. Other than that, *Acori graminei*, which was found to be rich in β-asarone, enhanced cognitive function of AβPP/PS1 mice and decreased neuronal apoptosis in the AβPP/PS1 mouse cortex. In addition, a substantial increase in the expression of CaMKII/CREB/Bcl-2 was observed in the cortex of AβPP/PS1 mice treated with β-asarone.

In a study conducted by Ayuob et al. [113], *Ocimum basilicum* up-regulated the serum corticosterone level, the hippocampal protein glucocorticoid receptor, and the brain-derived neurotropic factor (BDNF); however, it down-regulated the neurodegenerative and atrophic changes induced in the hippocampus, which decreased after exposure to chronic unpredictable mild stress (CUMS). According to the data collected by Avetisyan and co-workers [216], the major components of *O. basil* includes methyl chavicol and linalool. Interestingly, many linalool-producing plants are commonly used in folk medicine and aromatherapy to alleviate symptoms and treat multiple acute and chronic diseases [217,218]. Linalool is frequently used in the manufacture of fragrances for shampoos, soaps, detergents, and in pharmaceutical formulations [219,220]. Research conducted by Sabogal-Guáqueta et al. [221] found that oral administration of monoterpene linalool to elderly mice (21–24 months old) with a triple transgenic form of AD (3x Tg-AD) at 25 mg/kg for three months at an interval of 48 h resulted in enhanced learning and spatial memory and increased risk assessment activity in the elevated plus maze. Hippocampi and amygdalae from 3x Tg-AD linalool-treated mice also showed a large reduction in extracellular β-amyloidosis, tauopathy, astrogliosis, and microgliosis, as well as pro-inflammatory marker levels of p38 MAPK, NOS2, COX2 and IL-1β. Thus, linalool is suitable as an AD prevention candidate for pre-clinical studies. Based on the articles that we have selected, linalool is a major volatile component of EOs in a number of aromatic plant species, such as *L. angustifolia Mill.*, *M. officinalis L*., *R. officinalis L*., and *C. citrate DC*. The presence of linalool can also help to reduce the deposits of Aβ, based on a study conducted by Gradinariu et al. [114] where Aβ1–42-treated rats exhibited the following: a decrease in exploratory activity (crossing number); smaller percentage of time spent and fewer entries in the open arm in the elevated plus-maze test; increase in swim time; and decrease in the immobility time within the forced swimming test.

### 4.6. Current EOs on PD

In terms of PD, the current therapy in practice is applied as a combination of gold-standard dopaminergic reposition with 3-(3,4-dihydroxyphenyl)-l-alanine (l-dopa), along with other agents such as MAO-B, catechol *O*-methyltransferase (COMT) inhibitors, dopaminergic agonists, and cholinergic blockers [222]. However, the available treatments are subject to consequences of motor and non-motor side effects, which leads to poor efficacy in advanced stages of PD [144]. These arising phenomena are the main reasons that suggest and emphasise the necessity for the synthesis of anti-PD drugs that could delay the progression of neurodegeneration [144]. As mentioned in the results, PD studies included in this review comprise in vitro and in vivo, as well as combinations of in vitro with in vivo or ex vivo research. *Cinnamomum* sp., *Eryngium* sp., *Myrtus* sp., *Acorus* sp., *Eplingiella* sp., *Foeniculum* sp., *Pulicaria* sp., *Rosa* sp., *Zingiber* sp., and *Lavandula* sp. were among the identified EOs based on the respective included PD studies.

Four of the included studies on PD were based on in vitro approaches, with various EOs. The first study was focused on the evaluation of protective effects of EOs extracted from *Cinnamomum* sp. (*C. verum* and *C. cassia*) and cinnamaldehyde, in comparison to hydroalcoholic extracts using 6-OHDA-induced PC12 cytotoxicity as the representative model of PD [138]. *Cinnamomum* sp., or more commonly known as cinnamon, belongs to the Lauraceae family that is composed of almost 250 species and has been acknowledged for extensive health benefits [223,224]. Among the various species, *C. verum* and *C. cassia* were the two main species that have been widely applied for their medicinal and culinary applications, especially in Iran [225,226,227]. It is important to note that cinnamaldehyde represents one of the key components of both species, and EOs were reported to exhibit strong antioxidant properties [223]. The findings of this study indicated that 6-OHDA led to cell death, cell apoptosis, and suppression of the p44/42 pathway. On the whole, the study concluded that synergistic effects of cinnamaldehyde and EOs as well as other extract components could promote cinnamon’s roles as neuroprotective agents, specifically for PD treatment [138].

*Eryngium* sp. belong to the Apiaceae family, and are recognised for their EOs’ potentials in MAO inhibition [139]. MAO is available in two forms (A, B) where MAO-A inhibition is linked to antidepressant effects, while MAO-B is correlated with PD treatment [228,229]. An in vitro study conducted by Klein-Júnior et al. demonstrated the assessment of *Eryngium* sp. (*E. floribundum*: *EP*, *E. horridum*: *EH*, *E. pandanifolium*: *EP*, *E. eriophorum*: *EE* and *E. nudicaule*: *EN*) EOs for their MAO inhibitory effect. Intriguingly, the findings of this study indicated that MAO-A activity was not inhibited by any EOs, while EPEO and EHEO resulted in MAO-B inhibition. The literature search has also highlighted that PD patients usually represent elevated levels of MAO-B which arise due to gliosis, and hence contribute towards the collapse of the dopaminergic system [229]. Thus, this study puts forward the claim that *Eryngium* sp. could have potential applications as CNS bioactive secondary metabolites, particularly for neurodegenerative disease, in relevance to characteristics exhibited by EHEO [139].

Another important aspect that is usually associated with studies on PD is on α-Syn fibrillation. It is presumed that protein structural modifications resulting in amyloid fibril formation progresses towards neurodegenerative disorders [140]. However, the exact factors of α-Syn actions of misfolding and aggregation in the brain are still scarce. In addition, it should also be noted that preventive measures against α-Syn fibrillation are yet to be available; hence, the acceleration of the fibrillation process via certain factors should be avoided. Among some of the common factors are metal ions, small molecules, nanoparticles, and, particularly, toxins that could intensify the aggregation process [140,230,231,232,233]. The next two in vitro studies were performed by the same research team, where they highlighted the effects of 15 various Iranian EOs against α-Syn fibrillation [140,141].

Among all the 15 oils tested, it was shown that *M. communis* demonstrated potential benefits because it elevated the fibrillation in a concentration-dependent manner. However, it is necessary to understand that the major components of this oil are not responsible for the observed changes, suggesting complexity of both extract and synergistic effects of the available compounds, regardless of their amount [140]. In the second study, the investigation on *C. cyminum* EO signified the presence of cuminaldehyde as the major active compound that plays its role in the inhibition of α-Syn fibrillation. In addition, cytotoxicity assays on PC12 cells indicated the absence of toxic effects with cuminaldehyde treatment throughout α-Syn fibrillation [141].

Apart from in vitro studies, PD studies are also extensively performed under in vivo conditions. EOs of *Acorus* sp. cover two of the in vivo studies in this review. Both of these studies were conducted by the same research team, and focused on the regulation effect of β-asarone that was isolated from *A. tatarinowii Schott* on 6-OHDA-induced parkinsonian rats via two distinct endoplasmic reticulum (ER) stress pathways [142,143]. ER is known for its role in protein folding, where the build-up of protein unfolding/misfolding could initiate a phenomenon called ER stress that further activates the cellular process of unfolded protein response (UPR) [234]. ER stress has been noticed in a number of PD experimental models, and is also provoked by an increase in wild-type α-Syn [235,236,237]. Three main pathways that are categorised as UPR are inositol requiring enzyme 1 (IRE1), protein kinase RNA (PKR)-like ER kinase (PERK), and activating transcription factor 6 (ATF6) [238].

In general, GRP78 functions in the regulation of ER stress, where it binds the three proteins of the UPR pathway and maintains them as inactive when the cells are not exposed to stress conditions. However, under conditions of accumulated protein unfolding or misfolding, GRP78 will bind to the proteins and release them [239,240,241]. In terms of autophagy, the latest research has claimed that it could be induced due to ER stress [242,243]. Beclin-1 is known for its role in forming autophagosomes and is an essential part of the initial autophagy process. The pro-autophagic role of Beclin-1 could be inhibited via its reaction with Bcl-2, however this interaction is also subject to disruption caused by Bcl-2 phosphorylation that leads to Beclin-1 release and accelerates autophagy [244]. As per the second pathway study, it was proven that β-asarone leads to Beclin-1 downregulation, which highlights that Bcl-2 could possibly be the main linkage between autophagy and ER stress. The findings of both studies lead us to conclude that diminishing ER stress via β-asarone regulation is proven to be useful in the impairment of PD pathological progression [142,143].

Another study that also applies the use of 6-OHDA-induced PD mouse models focused on the investigation of the effects of zingerone and eugenol on dopamine concentration, behavioural changes, and antioxidant activities upon 6-OHDA administration and treatment of l-dopa [28]. Zingerone is extracted from the ginger root, while eugenol originates from cloves and was reported to be protective against 6-OHDA-induced depletion of striatal dopamine via increases in SOD activity and elevation of reduced glutathione (GSH) and L-Ascorbate (Asc) concentration, respectively [245,246]. Although these groups of researchers previously reported positive findings where pre-treatment with zingerone or eugenol inhibited 6-OHDA-induced dopamine depression by preventing lipid peroxidation, the current study, which involved post-treatment with similar compounds, resulted in contradictory findings, where dopamine decrease was more pronounced [28,246]. Despite the availability of other findings that propose the benefits of consuming these compounds, Kabuto and Yamanushi [28] suggested that intake of these specific substances upon the onset of PD symptoms should be more carefully monitored to prevent further injury aggravation.

In addition to studies among 6-OHDA-induced Parkinson’s rat models, the possibility to achieve positive effects of EOs when complexed with β-cyclodextrin (βCD) was evaluated by Filho and colleagues using reserpine-induced progressive models for PD in mice [144]. Cyclodextrins are cyclic oligosaccharides that could form host–guest complexes with hydrophobic molecules and were also reported to protect EOs from heat, evaporation, moisture, oxidation and light effects along with facilitating easy solubility [247,248,249]. Complexation effects of cyclodextrins with EOs were shown to be more prominent in exerting positive effects, especially in the treatment of chronic diseases, as published by several studies [250,251]. In this particular study, the same approach was applied using leaf EO extracted from *Eplingiella fruticosa* (EPL), where one of the key components is 1,8-cineole. *Eplingiella* sp. belongs to the Lamiaceae family, and was reported for its benefits as anti-inflammatory and antioxidant effects [252,253]. This research demonstrated and proved the hypothesis whereby both treated groups of EPL and EPL-βCD deferred reserpine effects on catalepsy time. However, this effect was noticed to be more remarkable with EPL-βCD treated mice groups.

Another study also applied induction with reserpine, with a different PD model of ovariectomized and non-ovariectomized rats [145]. Reserpine is known as an irreversible inhibitor of the vesicular monoamine transporter 2 (VMAT-2). The approach of reserpine injection to rats as a mode of PD model was proposed in response to its action on the depletion of monoamine and locomotor activities [254]. Ovariectomized rats are subjected to oestrogen deficits, similar to surgically menopaused women, where cognitive damage is highly possible [255]. Lower oestrogen levels are correlated with many side effects, such as mental disorders, memory defects, emotional issues, and other cognitive failures [256]. These incidences have led to much attraction towards phytoestrogens for its protective nature against certain diseases. Fennel plant (*Foeniculum vulgare*) is classified as in the Apiaceae family, and is known for its phytoestrogen compounds; it showed promising results in the treatment of cognitive disorders, such as dementia and AD [145,257,258,259]. The evaluation of this study indicated that protective oestrogen effects against neurodegenerative disorders were significantly decreased among reserpine-induced ovariectomized rats. Injection of reserpine resulted in a more remarkable observation on limb movement disorder among ovariectomized rats. Fennel treatment at various doses for both groups gave better results on the motor activity, which stressed the importance of oestrogens and phytoestrogens as a protective measure of dopaminergic neurons and improved PD symptoms [145].

Rotenone administration to rats, as induction of a PD model, is an alternative approach where it induces nigrostriatal dopaminergic neuron degeneration that is associated with α-Syn Lewy bodies [260]. Rotenone is an insecticide with high lipophilic nature, and is known to inhibit mitochondrial complex-1 along with causing oxidative stress [261,262]. This rotenone-induced model was reported in the study by Issa and colleagues, based on the neuroprotective effects of *Pulicaria undulata* EO in male Wistar rats [146]. *P. undulata* belongs to the Asteraceae family, which is commonly distributed in Asia, Europe, and North Africa [263]. From this study, it was shown that EO of *P. undulata* could exert its neuroprotective effects via anti-inflammatory and antioxidant properties. The mechanisms involved in neuroinflammation suppression include downregulation of induced nitric oxide synthase (iNOS) expression, followed by lower gene expression of α-Syn [146].

Compared to individual studies, there are also several approaches that examine combined effects that could incorporate in vitro, in vivo, and also ex vivo applications. One such attractive research is on the combined in vitro/in vivo evaluation of SHXW EO with 1-methyl-4-phenyl-1,2,3,6-tetrahydropyridine (MPTP)-induced PD mice and SH-SY5Y cell lines [147]. In this study, SHXW was a Chinese herbal formulation that consisted of 15 crude herbs called KSOP1009, composed of eight medicinal plants of different families (Hamamelidaceae, Myristicaceae, Umbelliferae, Santalaceae, Piperaceae, Myrtaceae, Typhaceae, and Lamiaceae). MPTP is known to cause fast degeneration of dopaminergic neurons, and as such, it was believed that the use of this specific model could assist in explaining certain aspects of PD disease mechanisms [147,264]. Positive findings of the study demonstrated that ingestion of KSOP1009 was successful in the protection of MPTP toxicity, where this could be correlated with dopamine reduction that also decreases ROS and restores mitochondrial roles [147].

In terms of in vivo and ex vivo combinatorial approaches, two articles were highlighted in this review, with several authors originating from the same team [148,149]. Both studies explored the approach of L-dopa induction of oxidative toxicity. l-dopa has been recognised as the most effective symptomatic treatment of PD for more than 30 years; however, toxicity issues that were raised via in vitro studies seem to be an unresolved challenge [148]. It was also mentioned that prolonged L-dopa treatment is often associated with side effects that often result in a delay in its administration [148]. Past studies have claimed that l-dopa therapy in combination with antioxidants could lessen the possible side effects. As such, efforts were taken to evaluate the effects; this research team investigated the combined effects of EOs from *Lavandula angustifolia*, *Rosa damascena*, vitamin C and Trolox in the initial study [148], followed by another study in 2019 on the combined effects of pre-treatment with *Rosa damascena* and vitamin C [149].

For the first study, the obtained results indicated that both EOs from herbal plants showed noticeable radical scavenging and antioxidant properties against l-dopa toxicity. Similarly, the second study also put forward equivalent claims with *R. damascena* characteristics where it was in parallel to vitamin C, and exhibited a significant role of rose oil in its interference against the acute oxidative toxicity of l-dopa [148,149]. Based on all the collective studies, it could be observed that PD treatments remain centralised among several parameters that include mainly α-Syn fibrillation, MAO-B, β-asarone regulation of ER stress pathways, toxicity-induced models with 6-OHDA, MPTP, l-dopa, reserpine, and rotenone with common animal models of rats and mice. Although the regulation mechanisms involved in each of the parameters may differ, the main focus remains towards effective and improved treatments for PD patients.

### 4.7. Current EOs on Other Neurodegenerative Diseases

Our review findings also revealed that several studies did not specify the studied disease, and instead examined general neurodegenerative disorders. The in vitro study conducted by Costa et al. [150] using *Lavandula pedunculata* subsp. *lusitanica (Chaytor)* Franco EO suggested the ability of *L. pedunculata* as a suitable choice to prevent neurodegenerative disease. A study performed by Elmann et al. [151] with *Pelargonium graveolens* EO showed that some constituents may depend on synergistic interactions to function. Geranium oil from *Pelargonium graveolens* inhibited nitric oxide (NO) production, as well as the expression of the proinflammatory enzymes cyclooxygenase-2 (COX-2) and induced nitric oxide synthase (iNOS) in primary cultures of activated microglial cells. The finding showed that none of the major constituents could inhibit NO production when examined at natural relative oil concentrations, with excellent inhibitory activity of citronellol at higher concentrations. The findings indicated that the presence of synergistic interactions between these components are of considerable importance. Thus, geranium oil can be useful in neurodegenerative disease prevention where neuroinflammation is part of pathophysiology [151].

EOs could also be applied as inhalation-based treatment, which could signify a natural way to heal one’s mind, body, and soul [265]. In Bagci et al.’s study [153], it was investigated whether inhalation of the *Anthriscus nemorosa* EO leads to behavioural changes that indicated significant memory improvement and exhibited both anxiolytic- and antidepressant-like effects in dual-treated rats. The results suggested that *A. nemorosa* EO inhalation can prevent scopolamine-induced memory impairment, anxiety, and depression [153].

In the in vivo research conducted by Satou et al. [118], mice were administered *Rosmarinus officinalis* EO (EORO) by inhalation and it was concluded that the rate of spontaneous alternation activity was significantly improved. The key components (1,8-cineole, α-pinene and β-pinene) were detected in the brain in a concentration-dependent manner upon EORO inhalation, which indicated its possible exerted effects. However, in several other interpretations, it was proposed that 1,8-cineole might require synergistic interaction effects. This was highlighted by Costa et al. [150], where α-pinene and 1,8-cineole from *Lavandula pedunculata* subsp. *Lusitanica* were shown to be effective cholinesterase inhibitors even at low concentrations and are likely to contribute to this behaviour. The synergistic and antagonistic interactions between certain terpenes that could result in combined effects should also be taken into consideration. Although this compound was also mentioned in a study conducted by Hritcu et al. [112], the study focused more on the positive effect of linalool in the improvement of spatial memory deficit in a scopolamine-induced dementia rat model instead of the 1,8-cineole effect on neurodegeneration. In addition, a study by Kaufmann and colleagues [104] signified the potential of myrtenal as an effective AChE inhibitor, compared to 1,8-cineole with EOs of *Artemisia annua L.* (Asteraceae) or *Glycyrrhiza glabra L.* (Fabaceae).

## 5. Conclusions

In conclusion, our review highlighted that EOs provide many other benefits apart from its common usage as fragrances and flavours, especially for its significant role in neurodegenerative disease therapy. Their roles in reducing disease severity are exerted by different mechanisms that vary respective of their origin. Although human studies were not obtained as part of our search, we strongly believe that the presence of essential components such as 1,8-cineole, carvacrol, or β-asarone could play significant roles in efforts towards the prevention and treatment of neurodegenerative disorders. Therefore, it is vital that the search for novel species of EOs are continued, to explore oils which could be applied towards the application of EO-based therapies or treatment strategies intended for age-related neurodegenerative disorders.

## Figures and Tables

**Figure 1 molecules-26-01107-f001:**
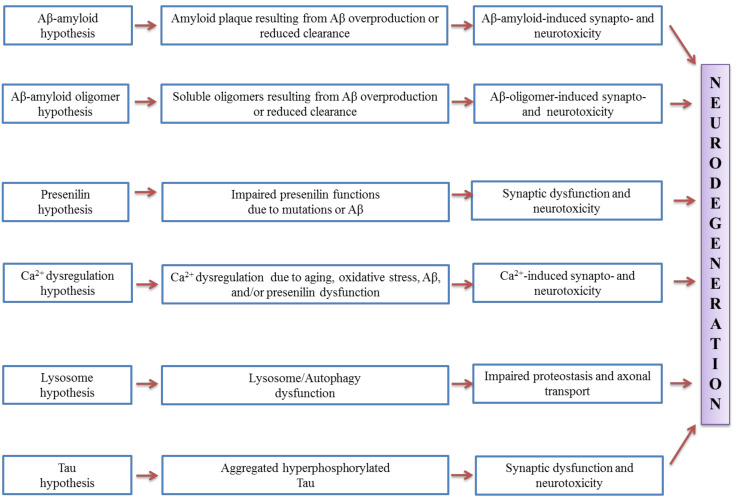
Pathogenic hypotheses for synaptic and neuronal toxicity in Alzheimer’s disease (adapted from [39]).

**Figure 2 molecules-26-01107-f002:**
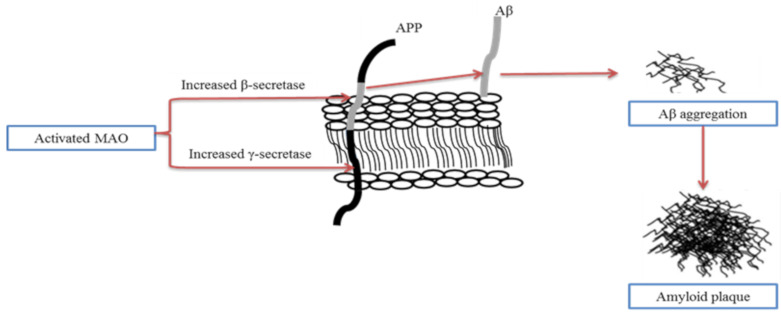
The mechanism of Aβ generation through modulation of amyloid precursor protein (APP) processing by activated monoamine oxidase (MAO). Activated MAO increases the expression of β-secretase and γ-secretase, improves Aβ generation, and contributes to the formation of amyloid plaques (adapted from [46]).

**Figure 3 molecules-26-01107-f003:**
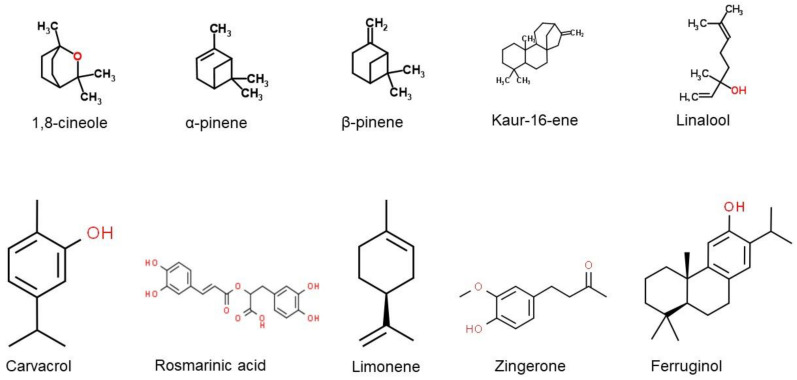
Chemical structures of the major EO components that have been reported to have anti-neurodegenerative properties. (obtained from [162,163,164,165,166,167,168,169,170,171]).

**Table 1 molecules-26-01107-t001:** Effects of essential oils (EOs) on neurodegenerative diseases.

No.	Disease	Study	Source of Essential Oils	Ref	Method	Findings	Conclusions and/or Recommendations
1.	**ALZHEIMER’S DISEASE**	In vitro	*Ajuga chamaecistus subsp scoparia (Boiss.)* Rech.f.	[64]	Chemical profiling [Gas chromatography (GC), Gas chromatography–mass spectrometry (GC–MS), High-performance liquid chromatography with diode-array detection (HPLC–DAD)] Cholinesterase inhibition assay (AChE, BChE—Ellman assay)	Major constituents in the EO—Spathulenol (18.0%), thymol (15.1%), octen-3-ol (14.3%), and linalool oxide (11.2%). Inhibit the activity of AChE (1.96 mg gallic acid equivalents/g), BChE (2.2 mg gallic acid equivalents/g).	This study should be extended to individual constituents of the EO and extracts of *A. chamaecistus.*
2.			*Allium tuncelianum* (Amaryllidaceae)	[65]	Chemical profiling [Gas chromatography–flame ionisation detector (GC–FID), GC–MS)] Anti-lipid peroxidation activity Cholinesterase inhibition assay (AChE, BChE—Ellman assay)	Major components—Diallyl disulfide (49.8%), diallyl trisulfide (27.9%) and allyl methyl trisulfide (6.9%). EO inhibition against BChE (28.65 ± 2.58%) and AChE (7.59 ± 2.90%) at 100 µg/mL.	*A. tuncelianum* could be a novel potency source of native antioxidant and anticholinesterase components.
3.			*Aloysia citrodora Palau*	[66]	Chemical profiling (GC–MS) Radioligand binding profile MTT cell viability assay Treatment of cell cultures with Aβ CytoTox 96 nonradioactive assay	The major chemical components—limonene, geranial, neral, 1, 8-cineole, curcumene, spathulenol and caryophyllene oxide. *A. citrodora* leaf EO inhibited [3H] nicotine binding to well-washed rat forebrain membranes, and increased iron-chelation in vitro. *A. citrodora* EO displays effective antioxidant, radical-scavenging activities and significant protective properties vs both hydrogen peroxide- and β-amyloid-induced neurotoxicity.	*A. citrodora* EO possess nicotinic cholinergic, oxidative and significant neuroprotective activities. These activities are of relevance to potential AD therapy, as well as other.
4.	**ALZHEIMER’S DISEASE**	In vitro	*Artemisia macrocephala*	[26]	Chemical profiling (GC–MS) Cholinesterase inhibition assay (AChE, BChE—Ellman assay)	The major components were α-humulene (46.3%), β-caryophyllene (9.3%), α-copaene (8.2%), β-myrcene (4.3%), *Z*(*E*)-α-farnesene (3.7%), and calarene (3.5%). Diverse volatile organic constituents (VOCs) of *A. maderaspatana* have significant AChE inhibitory activity (IC_50_ value = 31.33 ± 1.03 mg/mL).	The results of this study confirm the beneficial use.
5.			*Artemisia maderaspatana*	[67]	Chemical profiling (GC–FID, GC–MS) Cholinesterase inhibition assay (AChE—Ellman assay)	The major components were α-humulene (46.3%), β-caryophyllene (9.3%), α-copaene (8.2%), β-myrcene (4.3%), *Z*(*E*)-a-farnesene (3.7%), and calarene (3.5%). The experimental results showed that diverse VOCs of *A. maderaspatana* have significant AChE inhibitory activity (IC_50_ value of 31.33 ± 1.03 mg/mL).	The possibility of novel AChE inhibitors might exist in VOCs of this plant.
6.			*Artemisia absinthium*	[27]	Niosomal vesicles employing phosphatidylcholine, span 60, cholesterol and distearoylphosphatidylethanolamine with covalently linked polyethylene glycol of molecular weight 2000 (DSPE-PEG2000) by the thin-film method. *A. absinthium* was loaded into the niosomes. Inhibitory effect on aggregation of Aβ peptides using Thioflavin T fluorescence and atomic force microscopy.	Niosomes containing *A. absinthium* have a size of 174 ± 2.56 nm, the encapsulation efficiency of 66.73%, zeta potential of −26.5 ± 1/42 mV and polydispersity index (PDI) of 0.373 ± 0/02.	The nano-niosomes containing EO from art *A. absinthium* has the capability to preclude amyloid development.
7.	**ALZHEIMER’S DISEASE**	In vitro	*Boswellia dalzielii*	[68]	Chemical profiling (GC–FID, GC–MS) Cholinesterase inhibition assay (AChE—Ellman assay) Anti-inflammatory activity (5-lipoxygenase (LOX) inhibition assay)	Fifty compounds were identified, including 3-carene (27.72%) and α-pinene (15.18%). 2,5-Dihydroxy acetophenone and β-d-xylopyranose were identified in the methanol extract. IC_50_ = 35.10 mg/L for antixanthine oxidase and 28.01 mg/L for anti-5-lipoxygenase. For AChE inhibition, the best IC_50_ (76.20 and 67.10 mg/L) were observed, respectively, with an ethyl acetate extract and the EO. At 50 mg/L, the dichloromethane extract inhibited OVCAR-3 cell lines by 65.10%, while cyclohexane extract inhibited IGROV-1 cell lines by 92.60%.	The ethyl acetate and methanol extracts could be considered as potential alternatives for use in dietary supplements. They have a natural antioxidant, antihyperuricemic and anti-inflammatory activities.
8.			*Chamaecyparis obtusa* *Cryptomeria japonica*	[69]	Chemical profiling [(GC–MS, Nuclear magnetic resonance (NMR)] Cholinesterase inhibition assay (AChE, BChE)	Anti-AChE activity of *C. japonica* with 64.8% of inhibition at 100 μg/mL. The active principles of anti-AChE were investigated by activity-guided fractionation, and kaur-16-ene, nezukol and ferruginol were successfully identified, the IC_50_ values were 640, 300 and 95 µM, respectively. Kaur-16-ene and nezukol inhibited AChE in the mixed type mode, while ferruginol inhibited it in the competitive mode. In addition, nezukol and ferruginol showed anti-BChE activity, the IC_50_ values were 155 and 22 μM, respectively. Ethyl acetate extract of *C. obtusa* showed anti-AChE activity of 37.7% inhibition at 100 μg/mL. The active principle was determined to be (−)-hinokinin by activity-guided fractionation and the IC_50_ value was 176 μM.	The leaf of *C. japonica* and heartwood of *C. obtusa* may be suitable agents for AD therapy when administered through the nasal system as an aroma supplement.
9.			*Cinnamomum tamala* (Buch.-Ham.)	[18]	Chemical profiling [(High-performance liquid chromatography–photodiode array detection (HPLC-PDA)] Cholinesterase inhibition assay (AChE, BChE—Ellman assay and thin-layer chromatography (TLC) bio-autography)	HPLC analysis confirmed the presence of linalool in *C. tamala* EO. The cinnamon oil obtained from leaves of *C. tamala* possess maximum inhibition against AChE (IC_50_: 94.54 ± 0.774 µg/mL) and BChE (IC_50_: 135.56 ± 0.912 µg/mL). The result also explains that *C. tamala* is more sensitive to AChE enzyme than the BChE.	*C. tamala* may be explored as an anti-cholinesterase agent further for the better and safer management of ADs.
10.	**ALZHEIMER’S DISEASE**	In vitro	*Cinnamomum zeylanicum*	[19]	Chemical profiling (GC–FID, GC–MS) Monoamine oxidase (MAO)-A and MAO-B inhibitory activity (fluorometric assay) Cholinesterase inhibition assay (AChE, BChE—Ellman assay) Inhibition of self- and Cu^2+^-induced Aβ1–42 peptide aggregation (Thioflavin T-fluorometric assay) Tyrosinase inhibitory activity (L-dopa)	Twenty-two components were identified representing 99.79% of the oil. (*E*)-Cinnamaldehyde (CAL) (81.39%) and (*E*)-cinnamyl acetate (CAS) (4.20%) were determined as the major compounds. The oil and the major compounds, CAL and CAS, showed over 78.0% inhibitory activity on cholinesterases. In MAO A and MAO-B inhibition assay, both oil (96.44% and 95.96%, respectively) and CAL (96.32% and 96.29%, respectively) showed remarkable activity as high as rasagiline (97.42% and 97.38%, respectively). The oil also showed 57.78% and 84.53% inhibitory activity in self- and Cu^2+^ induced aggregation of Aβ1–42, respectively. In tyrosinase inhibition assay, CAL (83.75%) and CAS (45.58%) showed higher activity than that of the oil (18.08%).	Total oil exhibited remarkable anti-Alzheimer and skin whitening activity (over 80%).
11.			*Citrus Sinensis [L.] Osbeck*	[70]	Chemical profiling (GC–FID) Cholinesterase inhibition assay (AChE, BChE) MAO inhibition assay Lipid peroxidation assay	Forty-four compounds were identified in peels and seed EOs, respectively. The EOs inhibited AChE, BChE and MAO in a dose-dependent manner. Both EOs inhibited AChE activity. However, peel EO (IC_50_ = 2.64 μg/mL) had significantly (*p* < 0.05) higher inhibition than that of the seeds (IC_50_ = 3.54 μg/ mL). Similarly, the peel EO (IC_50_ of 2.61 μg/mL) had higher BChE inhibitory ability than seed EO (IC_50_ = 3.52 μg/ mL). MAO activity was inhibited compared to the reference test when the EO were added to the sample test. However, the EO from the seeds (IC_50_ = 6.93 μg/mL) had a significantly (*p* < 0.05) higher inhibitory ability compared to peel EO (IC_50_ = 8.53 μg/mL). In addition, EOs significantly (*p* < 0.05) inhibited malondialdehyde (MDA) produced in the brain homogenates to 168.21 and 141.47% for peel and seed EOs, respectively.	EOs from sweet orange peels and seeds inhibited cholinergic and monoaminergic enzymes and exhibited antioxidant.
12.	**ALZHEIMER’S DISEASE**	In vitro	*Citrus aurantifolia Swingle,* *C. aurantium L.* *C. bergamia Risso and Poit.*	[71]	Chemical profiling (GC–FID, GC–MS) Cholinesterase inhibition assay (AChE, BChE—spectrophotometric method)	The analysed EOs contain mainly limonene, α-pinene, β-pinene, γ -terpinene, and linalyl acetate. *C. aurantifolia* inhibited more selectively AChE. The best activity was exerted by *C. aurantifolia* and *C. aurantium* that inhibited AChE of EOs with IC_50_ values of 139.3 and 147.5 μg/mL, respectively. A lower bio-activity was observed against BChE with IC_50_ values ranging from 235.5 to 266.6 μg/mL for *C. aurantifolia* and *C. aurantium*, respectively.	Obtained data suggest a potential use of citrus oils as a valuable new flavor with functional properties for food or nutraceutical products with particular relevance to supplements for the elderly.
13.			*Cistus creticus* *Cistus salvifolius* *Cistus libanotis* *Cistus monspeliensis* *Cistus villo-sus*	[72]	Chemical profiling (GC–FID, GC–MS) Cholinesterase inhibition assay (AChE, BChE—Ellman assay)	*C. monspeliensis* exhibited the most promising activity in β-carotene bleaching test (IC_50_ of 54.7 μg/mL). *C. salvifolius* showed the highest activity against AChE (IC_50_ of 58.1 μg/mL) while *C. libanotis, C. creticus* and *C. salvifolius* demonstrated a good inhibitory activity against BChE with IC_50_ values of 23.7, 29.1 and 34.2 μg/mL, respectively.	Not possible to clearly demonstrate the relationship between phytochemicals and bioactivity further studies must be conducted.
14.			*Clinopodium serpyllifolium (M.Bieb.) Kuntze*	[73]	Chemical profiling (GC–MS) Cholinesterase inhibition assay (AChE, BChE—NA-FB)	The best AChE inhibitory activity was exhibited by the decoction extracts (IC_50_ = 2.17–2.36 mg/mL), followed by the MeOH extracts (IC_50_ = 3.03–3.12 mg/mL). The best BChE was exhibited by EO (IC_50_ = 2.86–3.10 mg/mL), followed by the decoction (IC_50_ = 3.31–3.87 mg/mL), and the MeOH extracts (IC_50_ = 4.43–4.66 mg/mL).	*C. serpyllifolium* could be a valuable natural source of antioxidants and cholinesterase inhibitors.
15.			*Daucus aristidis Coss.*	[74]	Chemical profiling (GC–MS) Cholinesterase inhibition assay (AChE, BChE—Ellman assay)	The main components of *D. aristidis* oils from Ghoufi were α-pinene (49–74.1%) and Bousaada were β-pinene (19.2–11.9%). The EO of *D. aristidis* from Boussaada region displayed modest inhibition against AChE and BChE (61.75% and 56.79%, respectively). The aerial parts, stems, leaves and umbels of *D. aristidis* EO from Ghoufi displayed low to moderate inhibition against AChE (51.0, 34.69, 13.44, and 33.07%, respectively) and BChE (41.46, 22.32, 23.70, and 30.00%, respectively).	EOs are complex mixtures of components that are usually more active than their isolated components. Their final activities are due to the combined effects of a number of minor components.
16.	**ALZHEIMER’S DISEASE**	In vitro	*Hedychium gardnerianum*,Sheppard ex Ker-Gawl	[75]	Chemical profiling (GC–MS) Cholinesterase inhibition assay (AChE—Ellman assay)	All the oils inhibited AChE, with IC_50_ values of approximately 1 mg/mL. Three oils presented mixed inhibition, whilst one was almost truly competitive.This activity can be attributed to presence of sesquiterpenes, which constituted more than 60% of the composition of the oils.	Oils may contribute to the increase in acetylcholine in cholinergic neurons and to the fight against deleterious oxidation.
17.			*Hymenocrater bituminous*	[76]	Chemical profiling (GC–MS) Cholinesterase inhibition assay (AChE, BChE)	GC–MS analysis of EO showed the presence of α-pinene (18.2%), β-pinene (11.3%), *trans*-phytol (11.0%), and spathulenol (8.5%) as the major components EO showed high AChE (3.8 mg GEs/g oil) and BChE (4.7 mg GEs/g oil) inhibitory activities.	The results indicated that *H. bituminous* has promising potential for possible uses in food and pharmaceutical industries due to its valuable phytoconstituents and biological activities.
18.			*Illicium verum* Hook.f	[77]	Chemical profiling (GC–MS) Cholinesterase inhibition assay (AChE, BChE—Ellman assay) Bioautography method for detection of AChE and BchE inhibition	Present study confirmed that anethole contributed to the anticholinesterase activity of *I. verum*, with more specificity towards AChE. IC_50_ for AChE and BChE inhibitory activity of anethole was 39.89 ± 0.32 μg/mL and 75.35 ± 1.47 μg/mL, whereas for the oil, 36.00 ± 0.44 μg/mL and 70.65 ± 0.96 μg/mL, respectively	*I. verum* fruit has potential inhibitory activity of AChE and BChE. This fruit and its oil could potentially lead to the development of anticholinesterase agents for the management of AD.
19.	**ALZHEIMER’S DISEASE**	In vitro	*Lavandula luisieri*	[78]	Chemical profiling (GC, GC–MS) Inhibitory activity on Beta-site APP-cleaving enzyme 1 (BACE-1). Endogenous BACE-1 in cultured cells	High contents of oxygen-containing monoterpenes, mainly necrodane derivatives, which are absent from any other oil. EO from *L. luisieri*, was signalised as inhibiting the BACE-1 enzyme. This oil was tested on the endogenous BACE-1 in cultured cells, being responsible for a reduction in Aβ production, with no significant toxicity. The main inhibitory activity was assigned to the monoterpenic ketone 2,3,4,4-tetramethyl-5-methylene-cyclopent-2-enone, one of the distinctive components of *L. luisieri* EO.	This inhibitor could be used to study functional mechanisms of BACE-1 or other aspartic proteases in cells or animal models. It could contribute to the research of disorders related to deregulation of these proteins.
20.			*Lavandula angustifolia*	[24]	Thioflavin T measurement Atomic force microscope (AFM) Imaging	The thioflavin T method showed that EO enhances the Aβ aggregation. The results of the AFM method also confirmed these effects.	Study suggests that combination of EOs (rose oil and lavender oil), vitamin C and Trolox with L-dopa can reduce oxidative toxicity, and may play a key role in ROS/reactive nitrogen species (RNS) disarming.
21.			*Lavandula pubescens*	[23]	Chemical profiling (GC–MS) Cholinesterase inhibition assay (AChE, BChE)	EO composition revealed 25 constituents, of which carvacrol (65.27%) was the most abundant. EO exhibited strong antiAChE (IC_50_ 0.9 μL/mL) and antiBChE (IC_50_ 6.82 μL/mL) effects.	LP EO makes a valuable natural source of bioactive molecules showing substantial potential as neuroprotective agents.
22.			*Lavandula viridis L’Her*	[79]	Chemical profiling (GC–FID, Gas Chromatography–Ion Trap Mass Spectrometry (GC–ITMS) Cholinesterase inhibition assay (AChE, BChE—Ellman assay) Qualitative determination of false-positive effects in AChE andBChE inhibition assay—TLC	Camphor was the main component identified in the EO (31.59 ± 1.32%), and in extracts from the first (1.61 ± 0.34%) and second supercritical fluid extraction (SFE) separators (22.48 ± 1.49%) at 12 MPa. The first separator SFE extract at 18 MPa was dominated by myrtenol (5.38 ± 2.04%) and camphor (4.81 ± 1.93%), second separator SFE extract was dominated by verbenone (13.97 ± 5.27%). Anti-cholinesterase activities EO was the most effective AChE inhibitor (IC_50_ = 411.33 ± 72.73 μg/mL) than first separator SFE extracts. First separator SFE extracts were the most effective BChE inhibitors (IC_50_ = 215.56 ± 13.60 and IC_50_ = 204.76 ± 22.86 μg/mL at extraction pressures of 12 and 18 MPa). The second separator extracts did not achieve 50% inhibition at the highest final concentration (2.5 mg/mL).	Phytochemicals from the aerial parts of *L. viridis* could be developed as natural antioxidant and anti-cholinesterase drugs, with particular applications in the symptomatic treatment of AD.
23.	**ALZHEIMER’S DISEASE**	In vitro	*Mentha spicata L.*	[80]	Chemical profiling (GC–MS) Cholinesterase inhibition assay: AChE, BChE—New micro-well plate AChE activity (NA-FB) method	Chemical analysis of EO revealed 31 compounds with oxygenated monoterpenes (90%) as the most abundant components followed by sesquiterpene and monoterpene hydrocarbons (6% and 3%, respectively). *M spicata* can be characterised as a carvone chemotype (65%). The EO of *M. spicata* showed high levels of inhibitory activity against AChE and BChE inhibited cholinesterase enzymes in a concentration-dependent manner, with AChE (IC_50_ = 23.1 μL/mL) and BuChE (IC_50_ = 35.0 μL/mL) inhibitory activities	The current study supports the utilisation of *M. spicata* EO as a traditional medicine and opens perceptions to find more potent substances in the EO for the management of obesity, AD, and dermatophytosis and for combating drug-resistant bacterial infections.
24.			*Neomitranthes obscura* (DC.) N. Silveira	[81]	Chemical profiling (GC–MS) Cholinesterase inhibition assay (AChE—Ellman assay)	17 compounds were identified, corresponding to 86.6% of the relative composition of this oil, being 3.9% of the oil constituted by monoterpenes and 82.7% constituted by sesquiterpenes. The major compounds found were *cis*-nerolidol, *trans*-nerolidol and β-bisabolene, corresponding to 19.3, 17.1, and 11.7% of relative composition, respectively. EO presented AChE inhibition with an IC_50_ of 75.93 ± 0.85 μg/mL.	EO from species of “Restinga de Jurubatiba” National Park could be used as anticholinesterase inhibitors. Neomitranthes obscura remains unexplored regarding its biological properties.
25.	**ALZHEIMER’S DISEASE**	In vitro	*Origanum rotundifolium Boiss.*	[82]	Chemical profiling (GC–FID, GC–MS) Cholinesterase inhibition assay (AChE, BChE—Ellman assay)	The major component was identified as carvacrol (56.8%) along with p-cymene (13.1%), (*Z*)-β-ocimene (5.4%), β-caryophyllene (3.9%), borneol (3.4%) and thymol (3.2%) Anticholinesterase activity against AChE with 38.66% inhibition, and BChE with 50.66% inhibition at 25 μg/mL concentration.	This is the first report on cholinesterase inhibitory activity of the EO of the plant.
26.			*Panax ginseng* *Panax japonicus* *P. notoginseng* *P. quinquefolius*	[83]	Chemical profiling (GC–MS) Cholinesterase inhibition assay (AChE, BChE) β-Secretase Inhibition	Spathulenol (8.82%), bicyclogermacrene (6.23%), β-elemene (3.94%), and α-humulene (3.69%) were identified as high content by GC–MS. EO (250 µg/mL) showed 41.4% inhibition against β-secretase, 77.4% inhibition against AChE, and 94.1% inhibition against BChE. Furthermore, β-elemene and α-humulene showed high activity among 3 compounds with 50% inhibitory concentration (IC_50_) values of 77.2 and 137.3 µM for AChE, and 298.2 and >2000 µM for BChE, respectively. An EO extract of *P. japonicus* showed the most potent activity with 51.3% inhibition at 500 μg/mL against β-secretase. *P. ginseng* showed the most potent inhibitory activity against AChE and BChE with 70.4% and 84.4% inhibition at 50 μg/mL, respectively. *P. notoginseng* extract showed the most potent activity with 57.3% inhibition at 500 μg/mL against Aβ aggregation.	An EO extract of *P. ginseng* could be an effective agent to improve AD symptoms.
27.			*Pinus sp.*	[84]	Chemical profiling (GC, GC–MS) Cholinesterase inhibition assay (AChE, BChE—Ellman assay)	Terpinolene, β-phellandrene, linalyl acetate, *trans*-caryophyllene, and terpinen-4-ol were identified *P. heldreichii* subsp. *leucodermis* exhibited the most promising activity, with IC_50_ values of 51.1 and 80.6 μg/mL against AChE and BChE, respectively. An activity against AChE was also observed with *P. nigra* subsp. *nigra* EO, with an IC_50_ value of 94.4 μg/mL. *Trans*-caryophyllene and terpinen-4-ol inhibited BChE with IC_50_ values of 78.6 and 107.6 μg/mL, respectively. β-Phellandrene was selective against AChE (IC_50_ value of 120.2 μg/mL).	Results demonstrated that *Pinus* EOs and some constituents have properties for the treatment of AD.
28.	**ALZHEIMER’S DISEASE**	In vitro	*Piper divaricatum*	[85]	Chemical profiling (GC–MS) Cholinesterase inhibition assay (AChE—TLC) Molecular docking	Methyl eugenol was the compound with the highest concentration ranging from 48.01% to 61.85%. The molecular docking revealed that β-elemene, eugenol, eugenyl acetate, and methyl eugenol are capable of interacting with different residues belonging to the active site of AChE, such as His447.	The results of per-residue free energy decomposition demonstrated that the molecules, during the simulation, performed interactions with residues of the active site that are important for the enzymatic activity inhibition.
29.			*Polygonum hydropiper L.*	[22]	Chemical profiling (GC–FID, GC–MS) Cholinesterase inhibition assay (AChE, BChE—Ellman assay)	141 and 122 compounds were identified in Ph.LO (leaf oils) and Ph.FO (flower oils), respectively. Caryophyllene oxide (41.42%) was the major component in Ph.FO while decahydronaphthalene (38.29%) was prominent in Ph.LO. In AChE inhibition, Ph.LO and Ph.FO exhibited 87.00 and 79.66% inhibitions at 1000 μg/mL with IC_50_ of 120 and 220 μg/mL, respectively. The IC_50_ value for galanthamine was 15 μg/mL. In BChE inhibitory assay, Ph.LO and Ph.FO caused 82.66 (IC_50_ 130 μg/mL) and 77.50% (IC_50_ 225 μg/mL) inhibitions, respectively, at 1000 μg/mL concentration.	Leaves and flowers of *P. hydropiper* exhibited dose-dependent anticholinesterase and antioxidant activities.
30.			*Prangos gaubae*	[86]	Chemical profiling (GC–MS) Cholinesterase inhibition assay (AChE, BChE)	EO analysis showed the presence of germacrene D (26.7%), caryophyllene oxide (14.3%), (*E*)-caryophyllene (13.8%), and spathulenol (11.3%) as the major volatile components. EO showed AChE (2.97 mg GEs/g oil) and BChE (3.30 mg GEs/g oil) inhibitory activities.	Results indicated that *P. gaubae* has promising potential for possible uses in food, cosmetic, and pharmaceutical industries.
31.	**ALZHEIMER’S DISEASE**	In vitro	*Rumex hastatus* D. Don	[87]	Chemical profiling (GC–FID, GC–MS) Cholinesterase inhibition assay (AChE, BChE—Ellman assay)	The GC–MS analysis of EO showed 123 components. Anticholinesterase assays demonstrated a marked potential against AChE and BChE with IC_50_ values of 32.54 and 97.38 μg/mL, respectively, which were comparable with the positive control, i.e., galantamine (AChE, IC_50_ = 4.73 μg/mL and BChE, IC_50_ = 11.09 μg/mL).	*R. hastatus* is an effective source of EO components having anticholinesterase and antioxidant potential.
32.			*Salvia officinalis*	[88]	Chemical profiling (GC–FID, GC–MS) Cholinesterase inhibition assay (AChE—Ellman assay) Anti-inflammatory activity (5-LOX inhibition assay)	Eighteen constituents were identified corresponding to 96.94% of present compounds. The main components were camphor (33.61%), 1,8-cineole (22.22%) and α-thujone (21.43%). EO inhibited the tested enzymes AChE (IC_50_ = 8.71 ± 2.09 mg/L), 5-LOX (IC_50_ = 36.15 ± 1.27 mg/L) and XOD (IP (%) = 36.89 ± 1.83 at a final concentration of 50 mg/L in the well.	*S. officinalis* EO can be considered as a source of bioactive phytochemicals and phytotherapeutics for humans due to its anti-Alzheimer activities.
33.			*Salvia chionantha*	[89]	Chemical profiling (GC–FID, GC–MS) Cholinesterase inhibition assay (AChE—Ellman assay)	Germacrene D (25.03%), β-caryophyllene (8.71%), spathulenol (5.86%) and α-humulene (4.82%) were identified as the major compounds. At 0.5 mg/mL concentration, the EO showed moderate AChE (56.7 ± 1.9%) and BChE (41.7 ± 2.9%) inhibitory activity.	The EO may be useful as a moderate anticholinesterase agent, particularly against AChE.
34.			*Salvia chrysophylla Staph*	[90]	Chemical profiling (GC–FID, GC–MS) Cholinesterase inhibition assay (AChE, BChE—Ellman assay)	The major components of the EO were α-terpinyl acetate (36.31%), β-caryophyllene (15.29%), linalool (8.12%) and β-elemene (4.26%). Anticholinesterase activity of the EO showed weak antioxidant activity. At 1 mg/mL concentration, the EO exhibited mild AChE (52.5 ± 2.0%) and moderate BChE (76.5 ± 2.7%) inhibitory activity.	Even if the EO demonstrated less AChE and BChE inhibitory activity than galantamine, it may be useful as a moderate BChE inhibitory agent. Some further studies should be done.
35.	**ALZHEIMER’S DISEASE**	In vitro	*Salvia urmiensis*	[20]	Chemical profiling (GC–FID, GC–MS) Cholinesterase inhibition assay (AChE, BChE—Ellman assay)	EO of leaves was rich in ester compounds such as ethyl linoleate (19%), methyl hexadecanoate (17%), and methyl linoleate (7.5%). The major compound of EO of flowers was 6,10,14-trimethyl-2-pentadecanone (55.7%). Exhibited moderate anticholinesterase activity (IC_50_ = 44–892 µg/mL). EO-F showed the highest AChE and BChE inhibitory potential (IC_50_ = 44 and 86 µg/mL, respectively) while EO-L showed inhibitory with IC_50_ of 211 and 422 µg/mL, respectively	The results indicated that *S. urmiensis* could be considered a valuable source for functional foods and pharmaceuticals.
36.			*Salvia nemorosa L.*	[91]	Chemical profiling (GC–FID, GC–MS) Cholinesterase inhibition assay (AChE—Ellman assay)	Sixteen components were identified in the oil extracted from flowers (91.3% of the total oil) and 8 components were identified in the oil extracted from leaves (94.0% of the total oil). Obtained oils from leaves and flowers contain similar major compounds which are spathulenol (57.8 and 23.0%) and caryophyllene oxide (28.2 and 45.0%, respectively). EO-L and EO-F are also dominated by oxygenated sesquiterpenes (86 and 68%, respectively). EO from *S. nemorosa* L.were generally less potent as an anti-AChE (IC 50 of EO-F: 488.9 ± 12.4 μg/mL and EO-L: 434.1 ± 11.6 μg/mL).	*S. nemorosa* maybe useful for novel applications in functional food and pharmaceutical industries
37.			*Salvia officinalis L.*	[21]	Chemical profiling (Fast GC–MS, Enantioselective GC–MS) Cholinesterase inhibition assay (AChE—Ellman assay) LOX inhibition assay	The main components were α-thujone (22.8–41.7%), camphor (10.7–19.8%), 1,8-cineole (4.7–15.6%) and β-thujone (6.1–15.6%). Enantioselective gas chromatography identified (−)-α-thujone and (+)-camphor as the main enantiomers in all the analysed EOs. EO from *S. officinalis* showed AChE inhibition with the following IC_50_ values expressed in μg/mL: SoEO-3 (326.7 ± 24.8) ≤ SoEO-1 (338.1 ± 13.8) ≤ SoEO-4 (450.0 ± 25.6) < SoEO-2 (867.4 ± 82.3). All four SoEO degrees of LOX inhibition (%) were as follows: SoEO-4 (52.8 ± 1.2) > SoEO-2 (45.5 ± 1.5) ≥ SoEO-1 (44.3 ± 1.3) ≥ SoEO-3 (42.0 ± 0.9).	The characterisation carried out increases our awareness of the possible uses of *S. officinalis* EO as natural additives in food, cosmetics and pharmaceuticals
38.	**ALZHEIMER’S DISEASE**	In vitro	*Salvia syriaca L.*	[92]	Chemical profiling (GC–MS) Cholinesterase inhibition assay (AChE, BChE—Ellman assay)	GC–MS analysis showed that spathulenol (87.4%), isospathulenol (7.6%), and bornyl acetate (2.7%) are the major compounds in EO. HPLC analysis indicated that rutin, quercetin, apigenin, rosmarinic acid, and ferulic acid are the most abundant phenolic components. The enzyme inhibitory potential of the tested EO and extracts in comparison with galantamine were weak (EO, AChE: 1.9 ± 0.1 mg/mL, BChE: 3.06 ± 0.1 mg/mL).	*S. syriaca* could be considered as a valuable source of bioactive natural compounds for functional foods, medical, and pharmaceutical applications.
39.			*Satureja thymbra L.*	[93]	Chemical profiling (GC–FID, GC–MS) Cholinesterase inhibition assay (AChE, BChE—Ellman assay)	Twenty-five compounds were identified by GC and GC–MS. The identified main compounds of the EO, carvacrol (34.6%), γ-terpinene (22.9%), p-cymene (13.0%) and thymol (12.8%) were also tested in the same manner. EO showed AChE (IC_50_: 150 ± 1.50 μg/mL) and BChE (IC_50_: 166 ± 2.00 μg/mL) inhibitory activities. The extract (IC_50_: 13.1 ± 0.23 μg/mL), the oil (IC_50_: 26.7 ± 0.56 μg/mL) and γ-terpinene (IC_50_: 11.9 ± 0.21 μg/mL) exhibited a good lipid peroxidation inhibitory activity.	Consumption of *S. thymbra* may protect people against oxidative stress and amnesia without any side effect.
40.			*Stachys inflata,* *S. lavandulifolia,* *S. byzantina*	[94]	Chemical profiling (GC–MS) Cholinesterase inhibition assay (AChE, BChE—Ellman assay)	The major volatile components of *S. inflata* were identified as germacrene D (21.6%) and β-pinene (15.6%). *S. lavandulifolia* contained mainly germacrene D (22.5%) and α-pinene (15.5%) whereas *S. byzantina* showed hexahydrofarnesyl acetone (25.7%) and valeranone (17.1%) as major volatile constituents. *S. inflata* EO had the highest activity with 7.68 and 5.20 mg GALAEs/g EO, *S. byzantina* EO were at the same level approximately at 6.62 and 5.06 mg GALAEs/g EO and *S. lavandulifolia* EO showed remarkable activity at 4.96 and 4.88 mg GALAEs/g EO against BChE and AChE, respectively.	EOs of hedgenettle (betony) plants could be employed in the preparation of formulations to be used in cosmetics, food, and pharmaceutical products due totheir valuable antioxidant, neuroprotective, hypoglycemic, anti-obesity, and skin-care effects.
41.	**ALZHEIMER’S DISEASE**	In vitro	*Sideritis galatica Bornm.*	[95]	Chemical profiling (GC–FID, GC–MS) Cholinesterase inhibition assay (AChE, BChE—Ellman assay)	23 components, representing 98.1% of *S. galatica* EO (SGEO) were identified. Monoterpene hydrocarbons (74.1%), especially α- (23.0%) and β-pinene (32.2%), were the main constituents in SGEO. The main sesquiterpene hydrocarbons were β-caryophyllene (16.9%), germacrene-d (1.2%) and caryophyllene oxide (1.2%), respectively. The cholinesterase inhibitory activity of SGEO was very low when compared to galantamine. AChE (IC_50_: 0.618 mg/mL) and BChE (IC_50_: 0.632 mg/mL) inhibition ability of SGEO appear to be close.	This investigation suggests that SGEO can be considered as a source of natural agents for the development of new natural products such as food additives and drugs.
42.			*S. ballsiana* *S. cyanescens* *S. divaricata* *S. hydrangea* *S. kronenburgii* *S. macrochlamys* *S. nydeggeri* *S. pachystachys* *S. pseudeuphratica* *S. rusellii*	[96]	Chemical profiling (GC–FID, GC– MS) Cholinesterase inhibition assay (AChE, BChE—Ellman assay)	*S. pseudeuphratica, S. hydrangea* and *S. divaricata* EOs demonstrated the most potent AChE inhibitory effect [50% inhibition concentration (IC_50_) = 26.00 ± 2.00 μg/mL, 40.0 ± 4.00, 64.68 ± 4.16, respectively]. The EO of *S. pseudeuphratica* demonstrated the highest inhibitory activity against AChE and BChE among the tested *Salvia* EOs.	Evidences from the study augment the importance of EOs obtained from *Salvia* sp. *Salvia* sp. could be used to treat Alzheimer’s disease.
43.			*Thymus mastichina L.*	[97]	Chemical profiling (fast GC–MS, enantioselective GC–MS) Cholinesterase inhibition assay (AChE—Ellman assay) 5- LOX inhibition assay	1,8-Cineole and linalool were the main components, followed by α-pinene, β-pinene and α-terpineol. (–)-Linalool, (+)-α-terpineol and (+)-α-pinene were the most abundant enantiomers. All four *T. mastichina* EO (α-pinene, β-pinene, limonene and 1,8-cineole) inhibited both lipoxygenase and AChE activities.1,8-cineole was the best AChE inhibitor with an IC_50_ of 35.2 ± 1.5 μg/mL. Bornyl acetate and limonene showed the highest lipoxygenase inhibition and 1,8-cineole was the best AChE inhibitor.	These results support the potential applications of TmEO as natural ingredients in nutracosmeceutical products.
44.	**ALZHEIMER’S DISEASE**	In vitro	*Thymus haussknechtii Velen.*	[98]	Chemical profiling (GC–FID, GC–MS) Cholinesterase inhibition assay (AChE, BChE—Ellman assay)	The major component of the EO was thymol (52.2%). *T. haussknechtii* showed anticholinesterase activity against AChE with 57.33% and BChE with 40.11% inhibition at 25 μg/mL concentration. The EO obtained from *T. haussknechtii* was rich in thymol (52.2%) and thymol exhibited strong acetyl- (83.0%) and BChE (98.0%) inhibitory activities.	These results indicate that *T. haussknechtii* could be a good source for natural antioxidants which were very important in prevention of many disease and protection of health.
45.			*Thymus lotocephalus*	[99]	Chemical profiling (GC–FID, GC–IT-MS) Cholinesterase inhibition assay (AChE, BChE—Ellman assay)	Linalool (10.43 ± 1.63%) was the main component in EO, whereas camphor (7.91 ± 0.84%) and *cis*-linalool oxide (7.25 ± 1.45%) were the major compounds in the extracts; second separator obtained at pressures of 12 and 18 MPa, respectively. Caryophyllene oxide was the primary constituent identified in the extracts; first separator (4.34 ± 0.51 and 4.41 ± 1.25% obtained at 12 and 18 MPa, respectively). The inhibitory effect produced by the EO and extracts-1st separator were clearly more pronounced than that of the extracts; second separator, which did not reach 50% inhibition of either enzymes at the highest concentration tested (2.5 mg/mL). The EO was significantly (*p* < 0.05) more active against BChE than AChE. Extraction pressure influenced the bioactivity of the SFE extract (first separator) isolated at the lowest pressure, which was the most active in inhibiting AChE (IC_50_ = 1.54 ± 0.04 mg/mL). BChE activity at the similar pressure indicates IC_50_ of 0.14 ± 0.02 mg/mL). BChE activity was more affected by EO than reference drug galanthamine, which was more active against AChE.	This research suggests *T. lotocephalus* as a source of bioactive compounds with different targets and pathways.
46.	**ALZHEIMER’S DISEASE**	In vitro	*Zingiber cassumunar*	[100]	Chemical profiling (GC–MS) Cholinesterase inhibition assay (AChE, BChE—Ellman assay)	IC_50_ values of native ZCEO show that it can be characterised as a moderate BChE inhibitor with IC_50_ of 0.355 ± 0.137 mg/mL and a weak AChE inhibitor with IC_50_ of 5.573 ± 0.176 mg/mL. Native ZCEO was formulated with microemulsion (ME) technique, and the suitable ZCEO ME was composed of Triton X-114 in combination with propylene glycol. The anticholinesterase activity of ME was higher than native ZCEO. It exhibited twenty times and twenty five times higher inhibitory activity against AChE (IC_50_ = 0.252 ± 0.096 mg/mL) and BChE (IC_50_ = 0.014 ± 0.002 mg/mL), respectively.	ZCEO loaded ME is an attractive formulation for further characterisation and an in vitro study in an animal model with AD.
47.			*Zosima absinthifolia Link*	[14]	Chemical profiling (GC–FID, GC–MS) Cholinesterase inhibition assay (AChE, BChE—Ellman assay) Anti-lipid peroxidation activity (TBA assay) Molecular docking	The GC–FID and GC–MS analysis revealed that the main components of the aerial parts, roots, flowers and fruits extracts were octanol (8.8%), octyl octanoate (7.6%), octyl acetate (7.3%); *trans*-pinocarvyl acetate (26.7%), β-pinene (8.9%); octyl acetate (19.9%), *trans*-*p*-menth-2-en-1-ol (4.6%); octyl acetate (81.6%), and (*Z*)-4-octenyl acetate (5.1%). The dichloromethane fraction of fruit had the best inhibition against BChE enzyme (82.27 ± 1.97%) which was higher than AChE inhibition (61.09 ± 4.46%) of umbelliferone. The highest antioxidant potential in the TBA assay was seen in the fruit CH_2_Cl_2_ fraction and flower EO (IC_50_ = 48.98 and 97.11 g/mL, respectively). The fruit CH_2_Cl_2_ fraction and flower EO indicated considerable inhibition against BChE (82.27 ± 1.97 and 78.65 ± 2.66%, respectively). The CH_2_Cl_2_ fractions of root and fruit also showed considerable inhibition against AChE (29.15 ± 2.45 and 31.46 ± 2.78%).	Study shows that the flowers and fruit of *Z. absinthifolia* can be a new potential resource of natural antioxidant and anticholinesterase compounds. Could be an herbal alternative to synthetic drugs in the prophylaxis of AD.
48.	**ALZHEIMER’S DISEASE**	In vitro	*Calamintha nepeta*, *Foeniculum vulgare*, *Mentha spicata* and *Thymus mastichina*	[101]	Chemical profiling Cholinesterase inhibition assay (AChE, BChE—Ellman assay)	The main components of EOs were oxygenated monoterpenes, and aqueous extracts were rich in phenol and flavonoid compounds. EOs (0.5 mg/mL) showed high ability to inhibit cholinesterase activities, with IC_50_ (concentration that inhibit 50% of enzyme activity) ranging from 0.08 to 0.24 and 0.09 to 0.23 mg/mL, for AChE and BChE, respectively.	EOs and aqueous extracts of *C. nepeta*, *T. mastichina*, *M. spicata* and *F. vulgare* from Alentejo region showed high antioxidant potential that may have an important role in the oxidative stress protection.
49.			*Cymbopogon citratus* (Gramineae), *Citrus hystrix* (Rutaceae) and *Zingiber cassumunar* (Zingiberaceae)	[102]	Cholinesterase inhibition assay (AChE, BChE—Ellman assay)	*C. citratus* oil exhibited the highest activity with IC_50_ values of 0.34 ± 0.07 µl/mL and 2.14 ± 0.18 µl/mL against BChE and AChE activity, respectively.	Citratus oil loaded microemusions are attractive systems for further in vivo studies in animal models with AD.
50.			*Alpinia galanga Linn., Centella asiatica Urban.,* *Cinnamomum bejolghota* (Buch. Ham.) Sweet, *Citrus aurantifolia* Swing, *Citrus hystrix DC.*, *Citrus maxima (Burm.) Merr., Citrus reticulata Blanco cv. Shogun, Citrus reticulata var. Fremont, Cymbopogon citratus Stapf.*, *Eupatorium odoratum Linn,. Melissa officinalis Linn., Ocimum basilicum Linn., Ocimum canum Sims., Ocimum gratissimum Linn., Ocimum sanctum Linn., Piper sarmentosum Roxb., Polygonum odoratum Lour., Polyscias fruticosa, Harms.Zingiber cassumunar Roxb,.* *Zingiber officinale Rosco*	[25]	Chemical profiling (GC–MS) Cholinesterase inhibition assay (AChE, BChE—Ellman assay)	GC–MS analysis revealed that the major constituents of *C. aurantifolia* leaf oil are monoterpenoids including limonene, l-camphor, citronellol, o-cymene and 1,8-cineole. EOs obtained from *Melissa officinalis* leaf and *Citrus aurantifolia* leaf showed high AChE and BChE co-inhibitory activities. *C. aurantifolia* leaf oil revealed in this study had an IC_50_ value on AChE and BChE of 139 ± 35 and 42 ± 5 μg/mL, respectively.	The excellent inhibitory activity of the EO of *C. aurantifolia* on both enzymes suggests that this oil could be a promising substance for prevention and treatment of AD. *C. aurantifolia* is an attractive natural source to inhibit anticholinesterase and therefore warrants further study. The effect of combination of natural EOs would also be an interesting issue for further studies.
51.	**ALZHEIMER’S DISEASE**	In vitro	*Lavandula angustifolia* *Coriandrum sativum*	[103]	Assessment of cell viability by water-soluble tetrazolium salt (WST)-8 Assessment of nuclear morphology Caspase-3 activity assay Assessment of intracellular reactive oxygen species (ROS) production	Lavender and coriander EOs (10 μg/mL) as well as linalool at the same concentration were able to improve viability and to reduce nuclear morphological abnormalities in cells treated with Aβ1–42 oligomers for 24 h. Lavender, coriander EOs and linalool were also showed to counteract the increase in intracellular ROS production and the activation of the pro-apoptotic enzyme caspase-3 induced by Aβ1–42 oligomers.	This study suggests that EOs and their main constituent linalool could be natural agents of therapeutic interest.
52.			*Artemisia annua L.* (Asteraceae) *Glycyrrhiza glabra L.* (Fabaceae)	[104]	Cholinesterase inhibition assay (AChE—Ellman assay)	1,8-cineole, carvacrol, myrtenal and verbenone apparently inhibited AChE; the highest inhibitory activity was observed for myrtenal (IC_50_ = 0.17 mM).	This is the first report to show the potential of myrtenal present in the EO of *A. annua L.* or *G. glabra L.* as an AChE inhibitor.
53.	**ALZHEIMER’S DISEASE**	In vitro	*Angelica sinensis* (Oliv) Diels. *Ligusticum Chuanxiong* Hort	[105]	Aβ25–35 and Aβ1–42 fibrils preparation 3-(4,5-dimethylthiazol-2-yl)-2,5-diphenyltetrazolium bromide (MTT) assay (SH-SY5Y and PC12 cells) Lactate dehydrogenase (LDH) release measurement Flow cytometric analysis Intracellular ROS measurement	Z-ligustilide (Z-LIG) is an essential oil originally isolated from umbelliferous plants. Z-LIG at 1–30 mM provided an effective neuroprotection, as evidenced by the increase in cell viability, as well as the decrease in LDH release and intracellular accumulation of reactive oxygen species. Z-LIG markedly blocked Aβ fibril-induced condensed nuclei and sub-G1 accumulation suggestive of apoptosis. Furthermore, Z-LIG substantially reversed the activation of phosphorylated p38 and the inhibition of phosphorylated Akt caused by Aβ25–35. Z-LIG effectively protects against fibrillar aggregates of Aβ25–35 and Aβ1–42-induced toxicity in SH-SY5Y and differentiated PC12 cells, possibly through the concurrent activation of PI3-K/Akt pathway and inhibition of p38 pathway.	The results taken together indicate that Z-LIG protects against Aβ fibrils-induced neurotoxicity possibly through the inhibition of p38 and activation of PI3-K/Akt signaling pathways concurrently. Z-LIG might be a potential candidate for further preclinical study aimed at the prevention and treatment of AD.
54.		In vivo	*Achillea biebersteinii* (Asteraceae)	[106]	Six groups of adult female Wistar rats (3 months old) were used. These are, control group, Sco-alone-treated group, diazepam alone-treated group, tramadol alone-treated group, Sco + *A. biebersteinii* EO 1% administrated group and Sco + *A. biebersteinii* EO 3% administrated groups. Memory performances were assessed by Y-maze task and radial-arm maze task. Anxiety and depressive-like behaviour were evaluated by elevated plus-maze and forced swimming tasks, respectively. Chemical profiling (GC–FID, GC–MS)	The scopolamine-only administered group showed impaired spatial memory as evidenced by a decrease in the spontaneous alternation percentage in the Y-maze test, and increase in the number of working and reference memory errors in the radial arm-maze test. In addition, scopolamine-only administered group displayed increased anxiety and depressive-like behaviour as evidenced by decrease in the exploratory activity, the percentage of the time spent and the number of entries in the open arm within elevated plus-maze test and decrease in swimming time and increase in immobility time within forced swimming test.	*A. biebersteinii* EO significantly improved memory formation, and reduced anxiety and depression-like behaviour in scopolamine-induced rats as compared to scopolamine-alone induced rats. The EO of this plant could be a good candidate for complementary therapy against neurological diseases such as AD.
55.	**ALZHEIMER’S DISEASE**	In vivo	*Boswellia serrata* (Burseraceae)	[107]	Mice were administered the 42-amino acid form of amyloid β-peptide (Aβ1–42) to induce AD and then treated with olibanum EO (OEO) at 150, 300, and 600 mg/kg, for two weeks. Olibanum, the resin of the *B. serrata* Following treatment, the AD mice were assessed by step-down test (SDT), dark avoidance test (DAT), and Morris water maze test (MWM). Blood and brain tissues were collected for biochemical assessments. Chemical profiling (GC–MS)	The main constituents of OEO were limonene, α-pinene, and 4-terpineol. Treatment with OEO prolonged t latency in SDT and DAT, but decreased error times. Escape latency decreased and crossing times rose in the MWM following OEO treatment (*p* < 0.5). Treatment with OEO also enhanced the acetylcholine levels and decreased the AChE levels in serum and brain tissue (*p* < 0.5). Additionally, OEO reduced amyloid plaques in the hippocampus and protected hippocampal neurons from damage. Furthermore, OEO decreased c-fos expression in hippocampus tissues from AD mice (*p* < 0.5).	OEO has a significant ameliorative effect AD-induced deterioration in learning and memory in AD mice. The mechanisms of these effects are related to the increase in acetylcholine.
56.			*Chamaecyparis obtuse*	[108]	To model AD, 4 μg of aggregated Aβ was injected into the hippocampus. To test the effects of EO of *C. obtuse* (EOCO), behavioural performance in the MWM was tested 4 days after injection. After behavioural testing, brain sections were prepared for 2,3,5-triphenyltetrazolium chloride (TTC) staining and terminal deoxynucleotidyl transferase dUTP nick end labeling (TUNEL) assay. Chemical profiling (GC–MS) AChE activity assays	Inhaled EOCO protected spatial learning and memory from the impairments induced by Aβ1–40 injection. In addition, the behavioural deficits accompanying Aβ1–40-induced AD were attenuated by inhalation of EOCO. Furthermore, AChE activity and neuronal apoptosis were significantly inhibited in rats treated with Aβ1–40 and EOCO compared to rats treated only with Aβ1–40. AChE activity was significantly elevated (0.46 ± 0.05 U/mg protein, *p* < 0.001) and AChE content was decreased markedly (18.54 ± 1.17 umole/mg protein, *p* < 0.001) in the Aβ1–40 group. However, after treatment with donepezil or EOCO, AChE activity was significantly decreased and AChE content was significantly increased compared with the Aβ1–40 group. In addition, the EOCO group had less of a reduction in AChE activity than the donepezil group. The EOCO was composed of 45 main compounds with significant differences in the contributions of major monoterpenes (67.97%) and sesquiterpenes (25.97%).	EOCO suppressed both AD-related neuronal cell apoptosis and AD-related dysfunction of the memory system. Thus, the results of this study support EOCO as a candidate drug for the treatment of AD.
57.	**ALZHEIMER’S DISEASE**	In vivo	*Citrus limon*	[109]	APP/PS1 double transgenic AD mice The effects of lemon EO (LEO) on learning and memory were examined using the MWM test, novel object recognition test, and correlative indicators, including a neurotransmitter (AChE), a nerve growth factor (brain-derived neurotrophic factor, BDNF), a postsynaptic marker (PSD95), and presynaptic markers (synapsin-1, and synaptophysin), in APP/PS1 mice. Histopathology was performed to estimate the effects of LEO on AD mice.	A significantly lowered brain AChE depression in APP/PS1 and wild-type C57BL/6L (WT) mice. PSD95/ Synaptophysin, the index of synaptic density, was noticeably improved in histopathologic changes. Hence, it can be summarised that memory-enhancing activity might be associated with a reduction in the AChE levels and is elevated by BDNF, PSD95, and synaptophysin through enhancing synaptic plasticity.	Offer in vivo evidence that LEO provides protection against AD-related synaptic loss and memory impairment. The finding also supports the importance of LEO as a “memory enhancer” in both the WT and APP/PS1 brain to mediate AChE, synaptic plasticity and cognitive function.
58.	**ALZHEIMER’S DISEASE**	In vivo	*Cymbopogon giganteus* *Illicium pachyphyllum* *Carum carvi*	[110]	Twenty-five rats were divided into five groups (/group): first group—control; second group— Aβ1–42 (1 mM) received donepezil treatment (5 mg/kg); third group— Aβ1–42 (1 mM); fourth and fifth groups—Aβ1–42 (1 mM) that received (−)-*cis*-carveol treatment groups (1% and 3%) Determination of the hippocampal AChE, SOD, CAT, Glutathione peroxidase (GPx) activity, Protein carbonyl and MDA level The total hippocampal content of reduced glutathione (GSH)	The results of this study demonstrated that (−)-cis-carveol improved Aβ1–42-induced memory deficits examined by using Y-maze and radial arm maze in vivo tests. Additionally, the biochemical analyses of the hippocampus homogenates showed that (−)-*cis*-carveol reduced hippocampal oxidative stress caused by Aβ1–42.	The present work suggested that (−)-cis-carveol provides neuroprotection against Aβ1-42 and can be regarded as an alternative therapeutic agent for dementia-related neurological conditions, including AD.
59.			*Ferulago angulata*	[111]	24 Male Wistar rats were divided into four groups: (1) Control group received 0.9% saline with 1% Tween 80 treatment; (2) Sco-alone received 0.9% saline with 1% Tween 80 treatment, as negative control; (3) Sco-treated received *F. angulata* EO 1%; and (4) Sco-treated received *F. angulata* EO 3%. Chemical profiling (GC–FID) Sco-induced memory impairments (Y-maze and radial arm-maze tasks). Decreased activities of superoxide dismutase, glutathione peroxidase and catalase; increase in AChE and decrease in total content of reduced glutathione	Production of protein carbonyl and malondialdehyde significantly increased in the rat hippocampal homogenates of scopolamine-treated animals as compared with control, as a consequence of impaired antioxidant enzymes activities. Additionally, in scopolamine-treated rats exposure to *F. angulata* EO significantly improved memory formation and decreased oxidative stress, suggesting memory-enhancing and antioxidant effects.	Our results suggest that multiple exposures to *F. angulata* EO amelioratescopolamine-induced spatial memory impairment by attenuation of the oxidative stress in the rat hippocampus.
60.	**ALZHEIMER’S DISEASE**	In vivo	*Lavandula angustifolia* ssp. angustifolia Mill. (Lamiaceae) *Lavandula hybrida* Rev. (Lamiaceae)	[112]	50 male Wistar rats were divided into 5 groups: (1) Control received saline treatment (0.9% NaCl); (2) Sco-treated received silexan, as + control; (3) Sco alone-treated; (4) Sco-treated received *Lavandula angustifolia* EO (LO1 + Sco); and (5) Sco-treated received *Lavandula hybrida* EO (LO2 + Sco). Chemical profiling (GC–MS) Behavioural recovery (maze task, forced swimming test) following chronic exposure to the extracted EO using sco-induced dementia rat model, are investigated.	The main components in both analysed samples, LO1 and LO2, were linalool (28.0% and 21.5%, respectively) and linalyl acetate (17%, 22.5%, respectively) followed by terpinen-4-ol (3.3% and 16.7%), lavandulyl acetate (8.3% and 8.4%). Interesting is that the presence of camphor and borneol was in trace amounts. Chronic exposures to lavender EOs (daily, for 7 continuous days) significantly reduced anxiety-like behaviour and inhibited depression in elevated plus-maze and forced swimming tests, suggesting anxiolytic and antidepressant activity. Additionally, spatial memory performance in Y-maze and radial arm-maze tasks was improved, suggesting positive effects on memory formation.	Multiple exposures to lavender EOs could effectively reverse spatial memory deficits in the rat brain and might provide an opportunity for management neurological abnormalities in dementia conditions.
61.	**ALZHEIMER’S DISEASE**	In vivo	*Ocimum basilicum*	[113]	Forty male Swiss albino mice divided into four groups (*n* = 10); the control, chronic unpredictable mild stress (CUMS), CUMS + Fluoxetine, CUMS + OB were used. Behavioural tests, serum corticosterone level, hippocampus protein level of the glucocorticoid receptors (GRs) and brain derived neurotropic factor (BDNF) were determined after exposure to CUMS. Hippocampus was histopathologically examined.	*O. basilicum* diminished the depression manifestation as well as impaired short term memory observed in the mice after exposure to the CUMS as evidenced by the forced swimming and elevated plus maze test. *O. basilicum* also up-regulated the serum corticosterone level, hippocampal protein level of the glucocorticoid receptor and the brain-derived neurotropic factor and reduced the neurodegenerative and atrophic changes induced in the hippocampus after exposure to CUMS.	EO of *O. basilicum* alleviated the memory impairment and hippocampal neurodegenerative changes induced by exposure to the chronic unpredictable stress indicating that it is the time to test its effectiveness on patients suffering from Alzheimer’s disease.
62.			*Ocimum sanctum L.* *Ocimum basilicum L.*	[114]	80 male Wistar rats (3 were divided into 8 groups: (1) Control received saline treatment (0.9% NaCl); (2) Diazepam alone-treated (DZP, 1.5 mg/kg); (3) Tramadol alone-treated (TRM, 10 mg/kg); (4) Aβ1–42 alone-treated; (5) Aβ1-42-treated received *O. sanctum* oil 1%; (6) Aβ1–42-treated received O. sanctum oil 3%; (7) Aβ1–42-treated received *O. basilicum* oil 1%; and (8) Aβ1–42-treated received *O. basilicum* oil 3%. Control, DZP, TRM and Aβ1-42 alone-treated groups were caged in the same conditions but in the absence of the tested oils. Chemical profiling (GC–MS) The anxiolytic- and antidepressant-like effects of inhaled basil EOs were studied by means of in vivo (elevated plus-maze and forced swimming tests) approaches.	The Aβ1–42-treated rats exhibited the following: decrease in the exploratory activity (number of crossing), the percentage of the time spent and the number of entries in the open arm within elevated plus-maze test and increase in the swimming time and decrease in the immobility time within forced swimming test. The main compounds found in both samples were linalool (31% Ob, 19% Os), camphor, β-elemene, α-bergamotene and bornyl-acetate, estragole (15.57 and 7.59%, respectively), eugenol (2.64 and 1.39%, respectively) and 1,8-cineole (3.29 and 3.90%, respectively). The exposure to basil EOs significantly improved the behaviour of the animals, suggesting anxiolytic- and antidepressant-like effects.	Multiple exposures to basil EOs can be useful to counteract anxiety and depression in AD conditions. Future perspectives of this study include a rather new cultivar of basil (*O. basilicum* var. *purpurascens*).
63.	**ALZHEIMER’S DISEASE**	In vivo	*Pimpinella peregrina*	[115]	Y-maze and radial arm-maze tests were used for assessing memory processes. Additionally, the anxiety and depressive responses were studied by means of the elevated plus-maze and forced swimming tests Chemical profiling (GC–MS).	The scopolamine alone-treated rats exhibited the following: decrease in the spontaneous alternation percentage in Y-maze test, increase in the number of working and reference memory errors in radial arm-maze test, along with decrease in the exploratory activity, the percentage of the time spent and the number of entries in the open arm within elevated plus-maze test and decrease in swimming time and increase in immobility time within forced swimming test. Inhalation of the *P. peregrina* EO significantly improved memory formation and exhibited anxiolytic- and antidepressant-like effects in scopolamine-treated rats.	Results suggest that the *P. peregrina* EO inhalation ameliorates scopolamine-induced memory impairment, anxiety, and depression. studies on the *P. peregrina* EO may open a new therapeutic window for the prevention of neurological abnormalities closely related to AD.
64.			*Pinus halepensis*	[116]	Rats were behaviourally tested (radial arm maze and Y-maze activities used). Rats were divided into five groups: first group—vehicle, second group—Aβ1–42, the third and fourth group—*P. halepensis* EO (PNO) treatment groups (1% and 3%), and fifth group—donepezil group (as positive control, 5 mg/kg injected in Aβ1–42-treated rats). Additionally, biochemical estimations of the brain homogenates for AChE and oxidative stress biomarkers were carried out. Chemical profiling (GC–FID, GC–MS) Determination of hippocampal AChE, SOD, CAT, GPX activity Determination of hippocampal protein carbonyl and MDA level	The EO reversed the Aβ1–42-induced decreasing of the spontaneous alternation in the Y-maze test and the Aβ1–42-induced increasing of the working and reference memory errors in the radial arm maze test. The Aβ1–42-induced modification of the balance oxidant-antioxidant and AChE action in the hippocampus of the rat has been ameliorated using the EO. Most of the volatile components belong to the sesquiterpene group (54.14%), and monoterpenes represent the rest of them, and only 2.50% are diterpenes (mainly cembrene). Aβ1–42 administration resulted in a substantial increase in the hippocampus AChE action (*p* < 0.0001) in comparison with the vehicle group. PNO (1% and 3%) treated groups significantly decreased the elevated AChE action (*p* < 0.0001 for PNO1% and *p* < 0.0001 for PNO3%) as compared to Aβ1–42 rats. In comparison with the vehicle group, injection of the Aβ1–42 resulted in significant decrease in SOD (*p* < 0.0001), CAT (*p* < 0.0001), and GPX (*p* < 0.0001). GSH levels were decreased after Aβ1–42 treatment (*p* < 0.0001), while elevated protein carbonyl (*p* < 0.0001) and MDA (*p* < 0.0001) levels were noticed as compared to vehicle groups.	*P. halepensis* EO may be regarded as a therapeutic tool for attenuation of Aβ toxicity and neuronal dysfunction.
65.	**ALZHEIMER’S DISEASE**	In vivo	*Kushui rose (Rosa setate × Rosa rugosa)*	[117]	Paralysis assay, Fluorescence staining of Aβ deposits assay, Exogenous rerotonin sensitivity assay, RNA interference (RNAi) assay, Subcellular DAF-16 or skinhead-1 (SKN-1) nuclear localisation assay, The gst-4::gfp expression assay, Western blotting	Rose EO (REO) significantly inhibited AD-like symptoms of worm paralysis and hypersensitivity to exogenous 5-HT in a dose-dependent manner. Its main components of β-citronellol and geraniol acted less effectively than the oil itself. REO significantly suppressed Aβ deposits and reduced the Aβ oligomers to alleviate the toxicity induced by Aβ overexpression. Additionally, the inhibitory effects of REO on worm paralysis phenotype were abrogated only after skn-1 RNAi but not daf-16 and hsf-1 RNAi. REO markedly activated the expression of gst-4 gene, which further supported SKN-1 signaling pathway was involved in the therapeutic effect of REO on AD *C. elegans.*	Our results provided direct evidence on REO for treating AD on an organism level and relative theoretical foundation for reshaping medicinal products of REO in the future.
66.	**ALZHEIMER’S DISEASE**	In vivo	*Rosmarinus officinalis*	[118]	Mice were administered Rosemary EO (EORO) by inhalation. Then, scopolamine was used to prepare Alzheimer’s type dementia model mice. To evaluate cognitive function, the Y-maze test was used for assessment of short-term memory. Chemical profiling (GC–MS)	EORO produced a significant improvement in the rate of spontaneous alternation behaviour. Furthermore, 1,8-cineole, α-pinene, and β-pinene, the main components of EORO, were detected in the brain in a concentration-dependent manner following inhalation of EORO. Components such as 1,8-cineole and others are likely involved in the effects on the brain.	The effect of improving cognitive function by inhaled administration of EORO, which has been used empirically, was clarified for the first time.
67.			*Salvia miltiorrhiza Bge*	[119]	Chemical profiling (GC–MS) The AD mice model was induced by D-gal plus AlCl_3_, and then the mice in the experimental group were treated with EO from aerial parts of *S. miltiorrhiza*. The protective effects of EO on the memory impairment of mice were determined by the MWM test. Biochemical determinations: superoxide dismutase (SOD), catalase (CAT), (MDA), AChE kits were used).	Terpenoids was the main components of EO from aerial parts of *S. miltiorrhiza*, accounting for about 50.18% of the total EO, and β-caryophyllene (8.58%), 6,10,14-trimethyl-2-pentadecanone (7.97%), dihydro-neoprene (7.96%), germacrene D (6.37%) caryop, hyllene (4.22%) were the main and characteristic compositions of the EO. The EO, given orally, prevented cognitive impairment in AD mice induced by D-gal plus AlCl_3_. Compared to the model group, SOD activity, CAT activity and Ach content were found to be increased in test group mice, while AchE activity and MDA content were decreased.	All the above suggest that EO from aerial parts of *S. miltiorrhiza* improves AD-like symptoms in mice induced by d-gal and AlCl_3_, and has the potential to develop a new drug for the treatment of AD.
68.	**ALZHEIMER’S DISEASE**	In vivo	*Schisandra chinensis*	[120]	Male KM mice weighing 20–25 g. Behaviour tests include open-field behaviour, Y-maze and MWM tests. Biochemical analysis include Western blot, Enzyme-linked immunosorbent assay (ELISA), immunofluorescence staining and histology assay. Cell viability assay and cytotoxicity (MTT assay) of *S. chinensis* EO (SEO) are also evaluated along with measurement of nitric oxide level.	SEO improved the cognitive ability of mice with Aβ1-42 or LPS-induced AD and suppressed the production of tumor necrosis factor-α (TNF-α), interleukin-6 (IL-6), and interleukin-1β (IL-1β) in the hippocampus. Furthermore, SEO inhibited p38 activation, but had little effect on other signaling proteins in the MAPK family, such as extracellular signal-regulated kinase 1/2 (ERK1/2) and c-Jun N-terminal kinase 1/2 (JNK). The SEO and BV-2 microglia co-culture was performed to further confirm the anti-inflammatory activity of SEO. The data showed that SEO decreased nitric oxide (NO) levels in LPS-stimulated BV-2 microglia and significantly blocked LPS-induced MAPKs activation.	Study suggests that SEO produces anti-AD effects on AD mice by modulating neuroinflammation through the NF-κB/MAPK signaling pathway.
69.			SuHeXiang Wan (SHXW)	[121]	We evaluated the effects of a modified SHXW (called KSOP1009) intake on the AD-like phenotypes of Drosophila AD models, which express human Aβ1–42 in their developing eyes or neurons.	When the flies were kept on the media containing 5 μg/mL of KSOP1009 extract, Aβ1–42-induced eye degeneration, apoptosis, and the locomotive dysfunctions were strongly suppressed. However, Aβ1–42 fibril deposits in the Aβ1-42 overexpressing model were not affected by treatment with KSOP1009 extract. Conversely, KSOP1009 extract intake significantly suppressed the constitutive active form of hemipterous, a JNK activator, while it induced eye degeneration and JNK activation.	Study suggests KSOP1009 confers a therapeutic potential to AD-like pathology of Aβ1-42 overexpressing Drosophila model.
70.	**ALZHEIMER’S DISEASE**	In vivo	*Tetraclinis articulata*	[122]	Chemical profiling (GC–FID, GC–MS) *T. articulata* EO was administered by inhalation to male Wistar rats once daily for 15 min period at doses of 1% and 3% for 21 days after the intracerebroventricular administration of Aβ1–42 right-unilaterally to induce memory deficits. Spatial memory of rats was tested using Y-maze and radial arm maze tests. In vivo brain antioxidant and AChE inhibitory effect.	The GC–MS and GC–FID data showed that the EO has a high percent of monoterpene hydro-carbons. EO reversed the Aβ1–42-induced decreasing of the spontaneous alternation in the Y-maze test and the Aβ1–42-induced increasing of the working and reference memory errors in the radial arm maze test. Furthermore, the Aβ1–42-decreased the AChE activity and the oxidant-antioxidant status in the rat hippocampus was retrieved by the treatment with the EO. AChE activity in the hippocampal tissues of Aβ1–42 group was higher as compared to control group (*p* < 0.01), while treatment by EO decreased AChE activity in the hippocampus as compared to Aβ1–42 group (*p* < 0.0001 for TLO1% and *p* < 0.001 for TLO3%). Thus, TLO reduced the cholinergic deficits produced after Aβ1–42 administration resulting in enhanced nootropic effect in the Y-maze and radial arm maze tests.	The study demonstrates that the EO could be a potent pharmacological agent against dementia by modulating cholinergic activity and promoting antioxidant action in the rat hippocampus.
71.			*Zataria multiflora Boiss.*	[123]	Forty male adult rats were categorised into four groups and treated as follows: 1. Negative Control (NC): no treatment; 2. Sham control (sham): distilled water by Intracerebroventricular (ICV) injection; 3. The AD control (AD): Aβ1–42 by ICV injection; and 4. The *Z. multiflora* EO (ZMEO) group: Aβ1–42 by ICV injection and ZMEO at 100 μL/kg/d orally for 20 days. Chemical profiling (GC–MS) AChE activity assay.	After Congo red staining of the hippocampus, a relative decrease in amyloid deposits was observed in the ZMEO group. Moreover, rats showed better outcomes in MWM test, reduced hippocampal AChE activity, and higher BDNF content as compared with the AD group (*p* < 0.05). However, no significant changes in antioxidant status was observed (*p* > 0.05). Thymol and carvacrol were the major constituents each comprising more than 30% of the EO, followed by p-cymene with 9.5%. The AChE enzyme activity significantly increased in the AD group as compared with the NC or sham groups (*p* < 0.001). Rats in the ZMEO group showed a significantly decreased activity of the enzyme in hippocampal tissue as compared with the AD group (*p* < 0.001). Interestingly, no significant difference was observed between the ZMEO, and NC or sham groups (*p* > 0.05).	ZMEO improves spatial learning and memory of rats with AD as assessed by MWM test. These effects are associated with decreased concentrations of hippocampal tau protein and TNF-α.
72.	**ALZHEIMER’S DISEASE**	In vivo	*Z. multiflora Boiss.*	[124]	Thirty-five adult male Sprague Dawley rats were randomly divided into 5 groups control (intact rats); sham (normal saline); AD control; vehicle control (rats with AD that orally received tween 80, 5% (ZMEO vehicle) for 20 days) and experimental (rats with AD that orally received ZMEO 100 µl/kg/day for 20 days). AD was induced by bidirectional microinjection of Aβ1–42. Tau protein and TNF-α concentrations were measured by ELISA methods. Spatial cognitive and noncognitive behaviour were determined by the MWM test.	ZMEO significantly improved latency time, time spent in the target quarter and cognitive behaviour of rats with AD compared to control and sham groups (*p* < 0.05). Hippocampal tau protein and TNF-α concentrations were significantly higher in both AD control and vehicle groups compared to control and sham groups, respectively (*p* < 0.01 and *p* < 0.001), administration of ZMEO reduced these parameters as compared to AD control and vehicle groups, respectively (*p* < 0.01 and *p* < 0.001).	ZMEO improves spatial learning and memory of rats with AD as assessed by MWM test. These effects are associated with decreased concentrations of hippocampal tau protein and TNF-α.
73.			*Origanum vulgare* *Thymus vulgaris*	[125]	Forty male albino rats divided into four groups as follows: control, AlCl_3_ induced AD, carvacrol oil treated and carvacrol nanoemulsion treated groups. Brain nor-epinephrine, serotonin and dopamine were analysed by HPLC. Levels of brain thiobarbituric acid-reactive substances (TBARS), SOD, GSH, cholinesterase, and advanced oxidation protein product (AOPP) were evaluated. Urinary 8-hydroxyguanosine (8-OHdG) level was evaluated by HPLC. Brain cyclooxygenase 1 and 2 (COX 1 and 2) were analysed by immunohistochemistry.	AD induced by AlCl_3_ in rats was depicted by the significant increase in the neurotransmitters levels which is accompanied with high degree of oxidative stress that was revealed in the elevated level of urinary 8-OHdG along with significant elevation in AOPP, TBARS, and cholinesterase levels and a significant decrease in SOD and GSH; these results are confirmed by immunohistochemistry analysis of COX 1 and 2. Treatment with carvacrol oil and carvacrol nanoemulsion were capable of mitigating effects mediated by AlCl_3_ administration in treated rats. While the treatment with both approaches succeeded to retract the negative impact of AlCl_3_; but the effect of carvacrol nanoemulsion was more notable than the EO. Carvacrol oil and carvacrol nanoemulsion were eminent to overturn AlCl_3_ induced brain AD which could be imputed to antioxidant and anti-inflammatory capabilities of carvacrol to alter oxidative stress effect. In extension; carvacrol nanoemulsion was evident to give a more effective and efficient way in carvacrol delivery to pass through blood–brain barriers and ameliorate brain changes.	Carvacrol nanoemulsion as a possible treatment for AD symptoms and its biochemical changes inside the brain is a new successful approach that combine between both.
74.	**ALZHEIMER’S DISEASE**	In vitro and In vivo	*Acori graminei*	[126]	Using PC12 cells and primary cultures of cortical neurons treated with Aβ1–40 or Aβ1–42 peptide. AβPP/PS1 mice at the age of 3 months and age-matched wild-type mice were intragastrically administered β-asarone (7 mg/kg/day, 21 mg/kg/day) or a vehicle daily for 4 months.	β-asarone can protect PC12 cells and cortical neurons and inhibit neuronal apoptosis by activating the CaMKII-α/p-CREB/Bcl-2 pathway. CaMKII-α overexpression enhanced the β-asarone-induced p-CREB-Bcl-2 expression and anti-apoptotic effects. Interestingly, suppression of CaMKII-α by siRNA or a specific inhibitor can significantly reduce the β-asarone-induced p-CREB and Bcl-2 expression and Aβ1–40 induced neuronal apoptosis in PC12 cells. β-asarone improved cognitive function of the AβPP/PS1 mice and reduced neuronal apoptosis in the cortex of the AβPP/PS1 mice. A significant increase in CaMKII/CREB/Bcl-2 expression was observed in the cortex of the AβPP/PS1 mice treated with β-asarone.	β-asarone can inhibit neuronal apoptosis via the CaMKII/CREB/Bcl-2 signaling pathway in in vitro models and in AβPP/PS1 mice. Therefore, β-asarone can be used as a potential therapeutic agent in the long-term treatment of AD.
75.	**ALZHEIMER’S DISEASE**	In vitro and In vivo	*Foeniculi vulgare aetheroleum*	[127]	Chemical profiling (GC–FID, GC–MS) Inhibition of 15-LOX Elevated plus-maze and forced swimming tests	*trans*-Anethole (58.1%), a phenylpropanoid, was found as the main component. Camphor (21.3%), a terpene ketone, was the second major compound detected in fennel EO, followed by carvone (7.8%), *trans*-sabinene hydrate (3.1%), β-pinene (2.6%), and others were found to be the minor components in the EO of fennel seeds. The Aβ1–42-treated rats exhibited the following: decrease in the exploratory activity, the percentage of the time spent and the number of entries in the open arm within elevated plus-maze test and decrease in swimming time and increase in immobility time within forced swimming test. Inhalation of the fennel EO significantly exhibited anxiolytic- and antidepressant-like effects. Our results suggest that the fennel EO inhalation ameliorates Aβ1–42-induced anxiety and depression in laboratory rats.	The results of the present study indicate that the fennel EO may have potential clinical applications in the management of anxiety and depression related to AD conditions.
76.			*Lavandula angustifolia Mill.*	[128]	Cognitive deficits were induced in C57BL/6J mice treated with sco (1 mg/kg, i.p.) and were assessed by MWM and step-through passive avoidance tests. Biochemical assays (AChE), SOD, GPx and MDA. In vitro, the cytotoxicity were induced by 4 h exposure to H_2_O_2_ in PC12 and evaluated by cell viability (MTT), lactate dehydrogenase (LDH) level, nitric oxide (NO) release, ROS) production and mitochondrial membrane potential (MMP).	LO (100 mg/kg) could improve the cognitive performance of scopolamine induced mice in behavioural tests. Meanwhile, it significantly decreased the AChE activity, MDA level, and increased SOD and GPx activities of the model. Moreover, LO (12 μg/mL) protected PC12 cells from H_2_O_2_-induced cytotoxicity by reducing LDH, NO release, intracellular ROS accumulation and MMP loss. AChE activity in the hippocampus of the model group was increased significantly compared with that of control mice (*p* < 0.05). The LO (100 mg/kg) and donepezil decreased the AChE activity in scopolamine treated mice (*p* < 0.05). Similarly, with the treatment of LO (100 mg/kg) and donepezil, significant attenuation of MDA level increase was observed in scopolamine treated mice (*p* < 0.05).	It was suggested that LO could show neuroprotective effect in AD model in vitro. LO may be a potential drug for the treatment of cognitive.
77.	**ALZHEIMER’S DISEASE**	In vitro and In vivo	*Lavandula angustifolia Mill.*	[129]	A total of 90 male C57BL/6J mice (12 weeks) were randomly divided into 6 groups, control (given vehicle), AD group (given vehicle), lavender EO (LO) + AD groups (50, 100 mg/kg/d), and linalool + AD groups (50, 100 mg/kg/d). Behavioural tests were performed using open-field test (OFT), MWM and passive avoidance task (PAT). Chemical profiling (GC–MS) AChE activity in the hippocampus and cortex were measured according to the protocol of assay kit	LO and linalool significantly protected the decreased activity of SOD, GPX, and protected the increased activity of AChE and content of MDA. Besides, they protected the suppressed nuclear factor erythroid 2-related factor 2 (Nrf2) and heme oxygenase-1 (HO-1) expression significantly. Moreover, the decreased expression of synapse plasticity-related proteins, calcium-calmodulin-dependent protein kinase II (CaMKII), p-CaMKII, BDNF, and TrkB in the hippocampus were increased with drug treatment. AChE activity in the hippocampus and cortex of the D-gal and AlCl_3_ group was increased significantly compared with that of the control group. The LO (100 mg/kg) and LI (100 mg/kg) protected the activity of AChE in the cortex of model mice markedly. Besides, linalool (100 mg/kg) also significantly reversed the D-gal and AlCl_3_-induced AChE activity increase in the hippocampus.	LO and its active component linalool have protected the oxidative stress, activity of cholinergic function and expression of proteins of Nrf2/HO-1 pathway, and synaptic plasticity. It suggests that LO, especially linalool, could be a potential agent for improving cognitive impairment in AD.
78.			*Lavandula luisieri*	[130]	β-Secretase and cathepsin D inhibition assays, cellular Aβ production inhibition assays, viability assays, sandwich ELISA, Detection of sAPPβ by Western blot	BACE-1 is an aspartic protease involved in the conversion of amyloid precursor protein (APP) to Aβ in vivo, which is one of the key steps in the development and progression of AD. In a previous screening procedure for inhibitors of BACE-1 activity, the oil of *L. luisieri* was identified as the most potent among several EOs. The overall results showed that compound 1 displayed a dose-dependent inhibition of BACE-1 in cellular and mouse models of AD and is therefore capable of passing through cellular membranes and the blood–brain barrier.	This inhibitor could be used to study functional mechanisms of BACE-1 or other aspartic proteases in cells or animal models. It could contribute to the research of disorders related to deregulation of these proteins.
79.	**ALZHEIMER’S DISEASE**	In vitro and In vivo	*Mentha longifolia*	[131]	Chemical profiling (GC–MS) Cholinesterase inhibition assay (AChE) In vivo anti-inflammatory test—Carrageenan induced pawoedema (Wistar rats)	The EO highest yield was recorded in the spring season. Pulegone (26.92%), 1.8 cineole (21.3%), and L-menthone (10.66%) were determined as its major compounds in the winter season. In the spring oil, the main components were pulegone (38.2%) and oleic and palmitic acids (23.79% and 15.26%, respectively). A remarkable AChE inhibitory activity of the ethyl acetate fraction (Ml EtOAcF) (IC_50_ = 12.3 μg/mL) and EO were also observed suggesting their neuroprotective property against AD.	It was found that its activity level was season-dependent. Further studies are still recommended to accurately identify the bioactive molecules in this extract.
80.			*Salviae aetheroleum*	[8]	Chemical profiling (GC–MS) We used in vitro and in vivo (the determination of SOD activity and AChE inhibitory activity) studies to fully assess its potential. For the in vivo testing, the EO was administered by inhalation to rats with induced AD and brain tissue samples were analysed.	The GC–MS analysis indicated the presence of 45 compounds and the principal components of the EO, in addition to thujone, were 1,8-cineole and camphor. The results of in vitro tests indicated that *S. aetheroleum* is a powerful inhibitor against 15-LOX (IC_50_ 0.064 µL/mL) and cholinesterase (IC_50_ 0.478 µL/mL) and a good scavenger of free radicals (IC_50_ 10.5 µL/mL). For the in vivo testing, daily exposures for one week to sage EO increased antioxidant enzymes activity, suggesting that the main mechanism to prevent neurodegeneration is related to antioxidant properties.	The EO extracted from dry leaves of *S. officinalis L.* has a significant antioxidant activity. It exhibited an anticholinesterase effect. It might be used for aromatherapy in patients with Alzheimer’s dementia in order to improve the oxidative status.
81.			SHXW	[132]	SHXW EO was extracted from nine herbs. The mouse AD model was induced by a single injection of Aβ1–42 into the hippocampus. The animals were divided into four groups, the negative control group injected with Aβ1–42, the Aβ group injected with Aβ1–42, the SHXW group inhaled SHXW EO and received Aβ1–42 injection, and the positive control group administered with docosahexaenoic acid (DHA,10 mg/kg) and with subsequent Aβ1–42 injection. Mice were analysed by behavioural tests and immunological examination in the hippocampus. In vitro investigation was performed to examine whether SHXW EO inhibits Aβ1–42 induced neurotoxicity in a human neuroblastoma cell line, SH-SY5Y cells.	Pre-inhalation of SHXW EO improved the Aβ1–42 induced memory impairment and suppressed Aβ1–42 induced JNK, p38 and Tau phosphorylation in the hippocampus. SHXW EO suppressed Aβ-induced apoptosis and ROS production via an up-regulation of HO-1 and Nrf2 expression in SH-SY5Y cells.	SHXW EO may have potential as a therapeutic inhalation drug for the prevention and treatment of AD.
82.	**ALZHEIMER’S DISEASE**	In vitro and In vivo	*Zelkova serrata*	[133]	Chemical profiling (GC–MS) Oxidative stress resistance assays in *C. elegans*, *C. elegans* paralysis assays	The EO of *Z. serrata* heartwood exhibited great radical scavenging activities and high total phenolic content. In vivo assays showed significant inhibition of oxidative damage in wild-type C. elegans under juglone induced oxidative stress and heat shock. Based on results from both in vitro and in vivo assays, the major compound in EO of heartwood, (−)-(1S, 4S)-7-hydroxycalamenene (1S, 4S-7HC), may contribute significantly to the observed antioxidant activity. Further evidence showed that 1S, 4S-7HC significantly delayed the paralysis phenotype in amyloid beta-expressing transgenic C. elegans.	These findings suggest that 1S, 4S-7HC from the EO of *Z. serrata* heartwood has potential as a source for antioxidant or AD treatment.
83.	**ALZHEIMER’S DISEASE**	In vitro and In vivo	*Zingiber officinale*	[134]	Phytochemical study was carried out using semi-preparative HPLC and GC–MS systems. Ginger methanolic extract (GME), the isolated pure compounds and ginger EO (GEO) were tested for their inhibiting activity in vitro against AChE using Ellman’s assay. The methanolic extract and the EO were studied in vivo using an AD model induced in rats using oral AlCl_3_.	GME and GEO showed moderate AChE inhibitory activity in vitro. On the other hand, the treatment with GME and GEO showed improvement in the learning and memory in Alzheimer’s model induced in rats also they showed significant inhibitory activity against AChE as compared to AD group (positive control group). Moreover, they showed an improvement of the morphological structure of the brain tissue with disappearance of most amyloid plaques. The preliminary screening of different concentrations (0.25, 0.50, 1 mg/mL) of GME and GEO showed inhibitory activity in a dose-dependent manner. However, at high concentration of 1 mg/mL, GME and GEO gave 41.4 ± 0.9% and 35.0 ± 1.9% inhibition of AChE, respectively	Study revealed the ability of GME and GEO to improve the symptoms of AD induced in rats. The inhibitory activity of both could be attributed to the presence of bioactive.
84.			*Pistacia khinjuk* *Allium sativum*	[135]	*P. khinjuk* EO(PKEO) and *A. sativum* EO (ASEO) were prepared and analysed in terms of extraction yield, phenolic content, and cholinergic markers in vitro. Moreover, both were administered orally to adult male Wistar rats at concentrations of 1, 2, and 3%. The inhibitory potential of PKEO and ASEO was compared with Donepezil (0.75 mg/kg) against the high activities of AChE and BChE enzymes.	PKEO reached an inhibition rate of 83.6% and 81.4% against AChE and BChE, respectively. ASEO had lower anti-cholinesterase activity (65.4% and 31.5% for the inhibition of AChE and BChE). PKEO was found to have more phenolic content than ASEO. A significantly positive correlation was observed between the total phenolics and anti-cholinesterase potential. In rats, both EO decreased the enzyme activity in a concentration-dependent manner. As compared with Donepezil, the significant difference in the AChE and BChE inhibition occurred as rats were treated with PKEO 3% (*p* < 0.05).	It could be concluded that PKEO and ASEO are potent inhibitors of AChE and BChE in rats that hold promise to be used for the treatment of AD. For future studies, it will be of interest to investigate their effect on oxidative stress in AD as one of the major cause of neurotoxicity.
85.	**ALZHEIMER’S DISEASE**	In vitro and Ex vivo	*Salvia* sp	[136]	The AChE inhibitory activity of different extracts from *S. trichoclada, S. verticillata*, and *S. fruticosa* was determined by the Ellman and isolated guinea pig ileum methods. The AChE inhibitory activity of the major molecule rosmarinic acid was determined by in silico docking and isolated guinea pig ileum methods.	The methanol extract of *S. trichoclada* showed the highest inhibition on AChE. The same extract and rosmarinic acid showed significant contraction responses on isolated guinea pig ileum. All the extracts and rosmarinic acid showed high radical scavenging capacities. Docking results of rosmarinic acid showed high affinity to the selected target, AChE.	In this study in vitro and ex vivo studies and in silico docking research of rosmarinic acid were used simultaneously for the first time. Rosmarinic acid showed promising results in all the methods tested. Our results suggest that rosmarinic acid may become a novel therapeutic candidate for the treatment of AD.
86.			*Syzygium aromaticum* *Xylopia aethiopica*	[137]	Chemical profiling (GC–MS) Cholinesterase inhibition assay (AChE, BChE—Ellman assay) TBARS—ex vivo	37 and 30 compounds were identified in the EOs from clove bud and Ethiopian pepper, respectively. High amounts of eugenol (85.61%) and β-caryophyllene (5.80%) were found in clove bud oils. Other minor components found in clove bud oils include α-caryophyllene (1.52%) and limonene (1.00%). The major components found in Ethiopian pepper, however, were eugenol (35.02%), terpinen-4-ol (7.21%), β-caryophyllene (5.98%), germacrene D (5.49%), (−) spathulenol (4.57%), and limonene (3.04%). The other minor components of the Ethiopian pepper oil include α-terpineol (2.71), α-caryophyllene (2.67%), β-phellandrene (2.67%), β-pinene (2.60%), eucalyptol (2.58%), β-linalool (2.28%) and α-pinene (1.75%). The EOs also inhibited AChE and BChE activities in a concentration-dependent manner (7.7–23.1 µL/L). Both oils were observed to have a stronger inhibitory effect on AChE than BChE. Furthermore, eugenol was identified as the major component in both EOs.	These results reveal the clove bud and Ethiopian pepper oils as potential sources of active metabolites with cholinesterase and antioxidant properties, although the Ethiopian pepper EO showed better potentials.
87.	**PARKINSON’S DISEASE**	In vitro	*Cinnamomum verum* *Cinnamomum cassia*	[138]	The cytotoxicity and cell apoptosis has been induced by 6-OHDA in PC12 cells. The protective effect was determined by measuring cell viability, the amount of reactive oxygen species (ROS), and apoptosis. Cell viability and apoptosis were assessed using resazurin assay, flow cytometry of propidium iodide (PI) stained cells, and Western blot analysis	6-OHDA resulted in the death and apoptosis of cells while, pretreatment with the extract and EOs of *C. verum* and *C. cassia* at 20 µg/mL and cinnamaldehyde at 5 and 10 µM for 24 h could significantly increase the viability (*p* < 0.001), and decrease ROS content (*p* < 0.05). Pretreatment with the extracts increased survivin and decreased cyt-c whereas pretreatment with the EOs decreased cyt-c, increased survivin, and reduced P-p44/42/p44/42 levels to a level near that of the related control. The extract and EOs of *C. verum* and *C. cassia* can be effective against 6-OHDA cytotoxicity.	It is suggested that, the synergistic efects of cinnamaldehyde and other components of extract and EOs promote cinnamon’s medicinal properties. Our study suggests that cinnamon components may be considered as neuroprotective agents in PD treatment.
88.			*Eryngium* sp.	[139]	Chemical profiling (GC–MS) MAO inhibition assay	EFEO, EEEO, ENEO, EHEO, and EPEO GC–MS analysis showed (E)-caryophyllene (4.9–10.8%), germacrene d (0.6–35.1%), bicyclogermacrene (10.4–17.2), spathulenol (0.4–36.0%), and globulol (1.4–18.6%) as main constituents. None of the EO inhibited MAO-A activity (4 and 40 µg/mL). However, EHEO inhibited MAO-B activity with an IC_50_ value of 5.65 mg/mL (1–200 mg/mL). Pentadecane (10 mM), its major constituent (53.5%), did not display significant MAOB inhibition.	The study demonstrates the promising application of Eryngium species as a source of potential central nervous system bioactive secondary metabolites, specially related to neurodegenerative disorders.
89.	**PARKINSON’S DISEASE**	In vitro	*Myrtus communis* *Ferula gummosa* *Eucalyptus globulus* *Satureja hortensis* *Rosmarinus officinalis* *Mentha spicata* *Mentha piperita* *Cuminum cyminum* *Artemisia dracunculus* *Citrus sinensis* *Citrus limonum* *Thymus vulgaris* *Lavandula officinalis* *Mentha pulegium* *Foeniculum vulgare*	[140]	The fibrillation process was monitored by thioflavin T fluorescence intensity. According to an increase in the permeabilisation of vesicles in the presence of the toxic aggregated species including oligomers and protofibrils, release of calcein from liposomes and the subsequent increase in the fluorescence signal due to dilution were measured. The MTT assay for cell viability evaluations.	*M. communis* particularly increased α-Syn fibrillation in a concentration-dependent manner. By using a unilamellar vesicle, it was shown that the aggregated species with tendency to perturb membrane were increased in the presence of *M. communis*. In this regard, the cytotoxicity of α-Syn on SH-SH5Y cells was also increased significantly. Inappropriately, the effects of fibrillation inhibitors, baicalein and cuminaldehyde, were modulated in the presence of *M. communis*. However, major components of *M. communis* did not induce fibrillation and also the effect of *M. communis* was limited on other fibrinogenic proteins.	Assuming that EOs have the ability to pass through the blood brain barrier along with the popular attention on aromatherapy for the incurable ND, these findings suggest an implementation of fibrillation tests for EOs.
90.			*Cuminum cyminum*	[141]	Site-directed mutagenesis of human α-SN and labeling with MBBr, Protein labeling with Fluorescein isothiocyanate (FITC) α-SN fibrillation process was monitored by Thioflavin T fluorescenceintensity Fractionation of *C. cyminum* (silica gel chromatography) and investigation the influences of the fractions on the α-SN fibril formation Chemical profiling (GC–MS) Discriminating apoptosis/necrosis using double stainingflow cytometric assay	Analysis of different fractions from the total extract identified cuminaldehyde as the active compound involved in the anti-fibrillation activity. In comparison with baicalein, a well-known inhibitor of α-SN fibrillation, cuminaldehyde showed the same activity in some aspects and a different activity on other parameters influencing α-SN fibrillation. The presence of spermidine, an α-SN fibrillation inducer, dominantly enforced the inhibitory effects of cuminaldehyde even more intensively than baicalein. Cuminaldehyde prevents α-SN fibrillation even in the presence of seeds, having no disaggregating impact on the preformed fibrils. Cytotoxicity assays on PC12 cells showed that cuminaldehyde is a nontoxic compound, treatment with cuminaldehyde throughout α-SN fibrillation showed no toxic effects on the cells.	Cuminaldehyde can modulate α-SN fibrillation. have potential therapeutic applications.
91.	**PARKINSON’S DISEASE**	In vivo	*Acorus tatarinowii Schott* (Shi Chang Pu)	[142]	To prove the hypothesis, we investigated the mRNA levels of glucose regulated protein 78 (GRP78) and C/EBP homologous binding protein (CHOP) in 6-hydroxydopamine (6-OHDA) induced parkinsonian rats after β-asarone treatment. Furthermore, the inositol-requiring enzyme 1/X-Box Binding Protein 1 (IRE1/XBP1) ER stress pathway was also studied.	β-asarone could improve the behaviour of parkinsonian rats; increase the HVA, Dopacl, and 5-HIAA levels; and reduce α-synuclein levels. Here, we assumed that the protective role of β-asarone on parkinsonian rats was mediated via the ER stress pathway. β-asarone inhibited the mRNA levels of GRP78 and CHOP, accompanied with the declined expressions of phosphorylated IER1 (p-IRE1) and XBP1. We deduced that β-asarone might have a protective effect on the 6-OHDA induced parkinsonian rats via IRE1/XBP1 Pathway	Collectively, all data indicated that β-asarone might be a potential candidate of medicine for clinical therapy of PD.
92.			*Acorus tatarinowii Schott* (Shi Chang Pu)	[143]	To support this hypothesis, we investigated the expressions of glucose regulated protein 78 (GRP78), PERK phosphorylation (p-PERK), C/EBP homologous binding protein (CHOP), Bcl-2 and Beclin-1 in 6-OHDA-induced parkinsonian rats after β-asarone treatment.	The results showed that the β-asarone group and PERK inhibitor group had lower levels of GRP78, p-PERK, CHOP and Beclin-1 while having higher levels of Bcl-2.	β-asarone might regulate the ER stress-autophagy via inhibition of the PERK/CHOP/Bcl-2/Beclin-1 pathway in 6-OHDA-induced parkinsonian rats.
93.	**PARKINSON’S DISEASE**	In vivo	*Eplingiella fruticosa*	[144]	Chemical profiling (GC–FID, GC–MS). We evaluated the effects of EO (EPL) and EPL complexed with β-cyclodextrin EPL-βCD (5 mg/kg, p.o. for 40 days) on male mice submitted to the progressive reserpine PD model. Behavioural evaluations, lipid peroxidation quantification and immunohistochemistry for tyrosine hydroxylase were conducted.	Phytochemical analysis showed the main constituents comprised β-caryophyllene, bicyclogermacrene and 1,8-cineole. EPL delayed the onset of catalepsy and decreased membrane lipid peroxides levels in the striatum. EPL-βCD also delayed the onset of catalepsy, reduced the frequency of oral dyskinesia, restored memory deficit, produced anxiolytic activity and protected against dopaminergic depletion in the striatum and substantia nigra pars compacta (SNpc).	These findings showed that EPL has a potential neuroprotective effect in a progressive PD animal model. Furthermore, EPL-βCD enhanced this protective effects, suggesting a novel therapeutic approach to ameliorate the symptoms of PD.
94.			*Foeniculum vulgare*	[145]	To create a model of Parkinson’s drug reserpine subcutaneously at a rate of 3 mg was used. Rats were surgically ovariectomized. Ten groups, each consisting of 8 rats, were used. The authors studied a total of 48 ovariectomized rats. EO of fennel (50, 100, 200 mg/kg) were used for treatment for 5 days. After the 5 days of behavioural and motor tests were performed, the amount of oestrogen was measured.	In this study, rats were ovariectomized and fennel EO doses (50, 100, 200 mg/kg) significantly increased amounts of oestrogen. The motor disorders of the treated groups were reduced. Furthermore, the non-ovariectomized groups were treated with EO of fennel. The amount of oestrogen increased and motor activity as well as behaviour had improved in the control groups.	The present study affirms that EO of fennel improve Parkinson’s in animal models. Further studies are needed to demonstrate the possible effects of EO of fennel on individuals with Parkinson’s and postmenopausal women.
95.	**PARKINSON’S DISEASE**	In vivo	*Pulicaria undulata*	[146]	Chemical profiling (GC–FID, GC–MS) The neuroprotective effects of three dose levels (50, 100, and 200 mg/kg) of *P. undulata* EO (PUEO) in rotenone-induced model in male Wistar rats were investigated.	Carvotanacetone was the major component (80.14%). The middle and high doses of PUEO attenuated rotenone-induced behavioural deficits besides, hindering the decrease in striatal dopamine and ATP levels, with partial retardation in rotenone-induced body weight loss. Biochemical assessments illustrated that PUEO mitigated rotenone-induced increment in striatal interleukin-1β (IL-1β), tumor necrosis factor-α (TNF-α), and inducible nitric oxide synthase (iNOS). The reduction in malondialdehyde and increase in glutathione striatal contents depicted its antioxidant potential. Molecular docking study of carvotanacetone might justify the observed normalisation of the elevated iNOS level induced after exposure to rotenone.	This is the first study indicating the ability of PUEO to protect rats against rotenone-induced PD via anti-inflammatory and antioxidant activities with the ability to reduce α-synuclein gene expression.
96.			*Zingiber officinale* *Syzygium aromaticum*	[28]	Dopamine (DA), 3,4-dihydroxyphenylacetic acid (DOPAC), and homovanillic acid (HVA) Levels, SOD activity, CAT activity, GPx activity, GSH and Ascorbic determination.	Daily oral administration of eugenol/zingerone and injection of L-dopa intraperitoneally for 4 weeks following a single 6-OHDA injection did not improve abnormal behaviours induced by L-dopa treatment. 6-OHDA reduced the DA level in the striatum; surprisingly, zingerone and eugenol enhanced the reduction in striatal DA and its metabolites. Zingerone decreased catalase activity, and increased glutathione peroxidase activity and the oxidised L-ascorbate level in the striatum.Previously reported findings showed that pre-treatment with zingerone or eugenol prevents 6-OHDA-induced DA depression by preventing lipid peroxidation. However, the present study shows that post-treatment with these substances enhanced the DA decrease. These substances had adverse effects dependent on the time of administration relative to model PD onset.	These results suggest that we should be wary of ingesting these spice elements after the onset of PD symptoms.
97.	**PARKINSON’S DISEASE**	In vitro and In vivo	*SHXW*	[147]	Cell toxicity, apoptosis, and ROS levels were analysed in the human neuroblastoma cell line SH-SY5Y. After that, changes in animal behaviour and tyrosine hydroxylase (TH) protein levels in the substantia nigra (SN) of 1-methyl-4-phenyl-1,2,3,6-tetrahydropyridine (MPTP)-injected mice were examined. Three different doses of KSOP1009 (30, 100, and 300 mg/kg, *n* = 8 for each group) were administered daily for 7 d before MPTP injection and 14 d after MPTP injection, totaling 21 d.	MPP+, the active metabolite of MPTP, decreased the viability of SH-SY5Y cells, whereas KSOP1009 alleviated MPP+-induced cytotoxicity. KSOP1009 (10 and 50 mg/mL) reduced MPP+-induced ROS generation compared with the control group. Treatment with 1 mM MPP+ increased the percentage of depolarised/live cells, whereas KSOP1009 intake at a dose of 10 mg/mL decreased the percentage of these cells. The mean latency to fall in the rotarod test was reduced in mice treated with MPTP compared with the control group. However, mice receiving three different doses of KSOP1009 performed better than MPTP-treated animals. MPTP-treated mice were more hesitant and took longer to traverse the balance beam than the control animals. In contrast, KSOP1009-treated mice performed significantly better than MPTP-treated mice. Furthermore, the KSOP1009-treated groups had a significantly higher number of TH-positive neurons in the lesioned SN and significantly higher expression of TH in the striatum than the MPTP-treated group. MPTP treatment strongly induced Jun-N-terminal kinase (JNK) activation, whereas KSOP1009 suppressed MPTP-induced JNK activation. In addition, KSOP1009 intake reversed the decrease in the phosphorylation levels of cAMP-response element-binding protein in the brain of MPTP-treated mice. KSOP1009 also restored the decrease in dopaminergic neurons and dopamine levels in the brain of MPTP-treated mice.	KSOP1009 protected mice against MPTP-induced toxicity by decreasing ROS formation and restoring mitochondrial function.
98.	**PARKINSON’S DISEASE**	In vivo and ex vivo	*Rosa damascena Mill.* *Lavandula angustifolia Mill,*	[148]	Male non-inbred albino mice (25–40 g) were divided into six groups. The groups undergoing combination therapy were pre-treated first for one hour with i.p. injections in doses of 400 mg/kg of ascorbic acid, Trolox, rose oil or lavender oils and received L-dopa and benserazide. The antioxidants protective effects against L-dopa oxidative by the levels of the three biomarkers of oxidative stress (OS)—MDA reactive substances, protein carbonyl content (PCC) and NO radicals in brain homogenate and blood of experimental mice.	Statistically significant increase relative to controls were observed in the combination rose oil + L-dopa (mean 3.06 ± 0.28 nmol/mL vs mean 1.51 ± 0.07 nmol/mL, *p* < 0.05, *t*-test). Similar results were seen in combinations of lavender oil + L-dopa, statistically significantly higher than controls (mean 3.74 ± 0.21 nmol/mL vs mean 1.51 ± 0.07 nmol/mL, *p* < 0.05, *t*-test) The levels of MDA in the combinations of rose oil + L-dopa were slightly higher vs controls (mean 2.11 ± 0.1 μmol/mL vs mean 1.88 μmol/mL ± 0.02, *p* < 0.1, *t*-test), and was statistically lower than the samples of mice treated with L-dopa (mean 2.11 ± 0.11 μmol/mL vs mean 2.87 ± 0.07 μmol/mL, *p* < 0.00, *t*-test). The combination lavender oil + L-dopa showed a statistically significant increase in comparison with the controls (mean 2.05 ± 0.15 μmol/mL vs mean 1.88 ± 0.02 μmol/mL, *p* < 0.00, *t*-test), and compared to samples treated only with L-dopa observed a statistically significant decrease (mean 2.05 ± 0.15 μmol/mL compared to an average of 2.87 ± 0.07 μmol mL, *p* < 0.3, *t*-test).	Study suggests that combination of EOs (rose oil and lavender oil), vitamin C and Trolox with L-dopa can reduce oxidative toxicity, and may play a key role in ROS/RNS disarming.
99.	**PARKINSON’S DISEASE**	In vivo and ex vivo	*Rosa Damascena Mill.*	[149]	Male non-inbred albino mice (25–40 g) and were divided into four groups. The control group of mice was inoculated two i.p. injections with solvent, only. The second injection was administered 45 min after the first. To study the L-dopa effect we used the acute model. The mice from all tested groups (except controls) received either two i.p. injections of L-dopa (100 mg/kg) followed by benserazide (10 mg/kg). The second injection was administered 45 min after the first. The groups undergoing combination therapy were pre-treated first for 1 h with i.p. injections with ascorbic acid (400 mg/kg), rose oil (400 mg/kg) and after that received L-dopa and benserazide. All mice were sacrificed by light anesthesia (Nembutal 50 mg/kg i.p.) after 30 min. Levels of lipid peroxidation measured as MDA, Protein carbonyl content (PCC), and advanced glycation end products (AGEs) in blood plasma of experimental model of healthy mice	Statistically significant increased MDA levels, PCC and AGEs were found in the blood L-dopa treated mice compared to the controls, while the same parameters were significantly decreased in the group pre-treated with antioxidants compared to the same controls. The MDA level in the combination L-dopa + ascorbic acid was statistically significantly higher than in controls (mean 2.08 ± 0.04 μmol/mL vs mean 1.77 ± 0.03 μmol/mL, *p* < 0.00, *t*-test) and statistically significantly lower than in mice treated with L-dopa alone (mean 2.08 ± 0.04 μmol/mL vs mean 3.75 ± 0.12 μmol/mL, *p* < 0.00, *t*-test). MDA level in combination L-dopa + Rose oil is statistically significantly higher than controls (mean 2.16 ± 0.05 μmol/mL vs mean 1.77 ± 0.03 μmol/mL, *p* < 0.00, *t*-test), and statistically lower than samples treated with L-dopa alone (mean 2.16 ± 0.05 μmol/mL vs mean 23.75 ± 0.12 μmol/mL, *p* < 0.00, *t*-test). Statistically significant increase in PCC is observed in group treated with L-dopa + Ascorbic acid versus controls (mean 3.18 ± 0.09 nmol/mg vs mean 1.55 ± 0.04 nmol /mg, *p* < 0.00, *t*-test), but compared with group treated with L-dopa alone, the PCC level was statistically significant lower (mean 3.18 ± 0.09 nmol/mg vs. 5.23 ± 0.01 nmol/mg, *p* < 0.00, *t*-test). The L-dopa + rose oil combination compared to controls was statistically significantly higher (mean 3.07 ± 0.02 nmol/mg vs. mean 1.55 ± 0.04 nmol/mg, *p* < 0.00, *t*-test), and statistically significantly lower in samples treated with L-dopa alone (mean 3.07 ± 0.02 nmol/mg vs. mean 5.23 ± 0.35 nmol/mg, *p* < 0.00, t-test). Ascorbic acid + L-dopa (686 ± 21.6 μg/mL, vs 432 ± 19.05 μg/mL, *p* <.00, *t*-test), and rose oil + L-dopa (489 ± 13.1 μg/mL, vs 432 ± 19.05 μg/mL, *p* < 0.000, *t*-test). Both groups pretreated with antioxidants showed a statistically significant decrease compared to group treated with L-dopa alone (*p* < 0.00).	The studied antioxidants can protect organisms from induced L-dopa oxidative toxicity and may play a key role in end products protection.
100.	**OTHER NEURODEGENERATIVE DISEASE**	In vitro	*Lavandula pedunculata subsp lusitanica* (Chaytor) Franco	[150]	Chemical profiling (GC–MS, HPLC-DAD) TBARS assay Cholinesterase inhibition assay (AChE, BChE—Ellman assay)	GC–MS analysis revealed that oxygen-containing monoterpenes was the principal group of compounds identified in the EO. Camphor (40.6%) and fenchone (38.0%) were found as the major constituents. HPLC–DAD analysis allowed the identification of hydroxycinnamic acids (3-O-caffeoylquinic, 4-O-caffeoylquinic, 5-O-caffeoylquinic and rosmarinic acids) and flavones (luteolin and apigenin) in the polar extracts, with rosmarinic acid being the main compound in most of them. The bioactive compounds from *L. pedunculata* polar extracts were the most efficient free-radical scavengers, Fe^2+^ chelators and inhibitors of malondialdehyde production, while the EO was the most active against AChE.	Subspecies of *L. pedunculata* is a potential source of active metabolites with a positive effect on human health. This research suggests *L. pedunculata subsp. lusitanica* as a source of natural compounds able to prevent neurodegenerative diseases.
101.			*Pelargonium graveolens*	[151]	Chemical profiling (GC–MS) Preparation of primary cultures of microglial cells, nitrite quantification, Western blot analysis	GC–MS analysis revealed presence of citronellol (26%), citronellyl formate (16%), linalool (10%), geraniol (8%), isomenthone (6%), and menthone (4%). Geranium oil inhibited NO production, as well as the expression of the proinflammatory enzymes cyclooxygenase-2 (COX-2) and induced nitric oxide synthase (iNOS) in primary cultures of activated microglial cells. When tested at natural relative concentrations in the oil, none of the major constituents of the oil could inhibit NO production. At higher concentrations, citronellol exhibited an inhibitory activity.	The results suggest a possible synergistic interaction between these components. Thus, geranium oil might be beneficial in the prevention/treatment of neurodegenerative diseases where neuroinflammation is part of the pathophysiology.
102.	**OTHER NEURODEGENERATIVE DISEASE**	In vitro	*Myrcianthes myrsinoides* (Kunth) *Grifo* *Myrcia mollis* (Kunth) DC. (Myrtaceae)	[152]	Chemical profiling (GC–FID, GC–MS), enantioselective gas chromatography, and gas chromatography–olfactometry (GC–O) Anticholinesterase activity (AChE, BChE)	A total of 58 compounds for *Myrcianthes myrsinoides* EO and 22 compounds for *Myrcia mollis* EO were identified and quantified by GC–MS with apolar and polar columns (including undetermined components). Major compounds (> 5.0%) were limonene (5.3–5.2%), 1,8-cineole (10.4–11.6%), (Z)-caryophyllene (16.6–16.8%), *trans*-calamenene (15.9–14.6%), and spathulenol (6.2–6.5%). The enantiomeric excess of eight chiral constituents was determined, being enantiomerically pure (+)-limonene and (+)-germacrene D. Eight components were identified as determinant in the aromatic profile: α-pinene, β-pinene, (+)-limonene, γ-terpinene, terpinolene, linalool, β-elemene and spathulenol. For *M. mollis*, the major compounds (> 5.0%) were α-pinene (29.2–27.7%), β-pinene (31.3–30.0%), myrcene (5.0–5.2%), 1,8-cineole (8.5–8.7%), and linalool (7.7–8.2%). The enantiomeric excess of five chiral constituents was determined, with (S)-α-pinene and (+)-germacrene D enantiomerically pure. Finally, the *M. myrsinoides* EO has an inhibitory activity for cholinesterase enzymes (IC_50_ of 78.6 µg/mL and 18.4 µg/mL vs. AChE and BChE, respectively.	*M. myrsinoides* and *M. mollis* have different chemical compositions. The similarity in the aroma extract dilution analysis evaluation could justify the similarity. This activity is of interest to treat AD.
103.	**OTHER NEURODEGENERATIVE DISEASE**	In vivo	*Anthriscus nemorosa*	[153]	*Anthriscus nemorosa* EO was administered by inhalation in the doses of 1% and 3% for 21 continuous days and scopolamine (0.7 mg/kg) was injected intraperitoneally 30 min before the behavioural testing. Y-maze and radial arm-maze tests were used for assessing memory processes. Additionally, the anxiety and depressive responses were studied by elevated plus-maze and forced swimming tests.	The scopolamine alone-treated rats exhibited the following: decrease the percentage of the spontaneous alternation in Y-maze test, increase the number of working and reference memory errors in radial arm-maze test, decrease in the exploratory activity, the percentage of the time spent and the number of entries in the open arm within elevated plus-maze test and decrease in swimming time and increase in immobility time within forced swimming test. However, dual scopolamine and *A. nemorosa* EO-treated rats showed significant improvement of memory formation and exhibited anxiolytic- and antidepressant-like effects in scopolamine-treated rats.	These results suggest that *A. nemorosa* EO inhalation can prevent scopolamine-induced memory impairment, anxiety and depression.

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
