# Peer review of "Essential Oils as a Potential Neuroprotective Remedy for Age-Related Neurodegenerative Diseases: A Review"

_molecules, 2021, doi:10.3390/molecules26041107_

Round 1

Reviewer 1 Report

The paper entitled “Essential Oils as a Potential Neuroprotective Remedy for Age-2 related Neurodegenerative Diseases: A Review” is based on latest information about essential oils studied towards their activity against dementia and neurodegenerative disorders. Authors described the issues very detailed what caused that the paper is very long and hard to read.  The following points have to be improved and corrected:

  1. Abstract: in accordance with Authors guideline, this part should include maximum 200 words hence I propose shortening the abstract.
  2. Introduction is too long. This part should include only introductory information about essential oils and basis of neurodegenerative disorders without the detailed information along with their mechanisms. If Authors want to present the detailed information, an additional part should be provide. Nevertheless, the presented mechanisms of neurodegeneration are commonly known and Authors should present only short information or the latest studies results. Detailed information can be presented in form of reference i.e.  https://doi.org/10.1016/j.biopha.2018.12.140 ; http://dx.doi.org/10.1016/j.biopha.2015.12.024
  3. Sections: Search strategy and Results are not necessary. Maybe should be removed?
  4. The Table 1. presents detailed information about essential oils, methods of studies and obtained results. Nevertheless, many of the information are duplicate in Discussion. I recommend improve these two parts. The shortening these parts allow to reduce whole paper what will make it more readable.
  5. Discussion: The information about MAO inhibition in neurodegeneration should be provide in AD description (see: point 2.)

On the whole, the paper is very interesting but in my opinion, Authors present too much detailed information what makes that the paper is hard to read. Please remove all reiterations and too much detailed mechanisms of action which should be basis for another review paper.  

Author Response

Point 1: Abstract: in accordance with Authors guideline, this part should include maximum 200 words hence I propose shortening the abstract.

Response 1: Abstract has been revised to less than 200 words.

Point 2: Introduction is too long. This part should include only introductory information about essential oils and basis of neurodegenerative disorders without the detailed information along with their mechanisms. If Authors want to present the detailed information, an additional part should be provide. Nevertheless, the presented mechanisms of neurodegeneration are commonly known and Authors should present only short information or the latest studies results. Detailed information can be presented in form of reference i.e.  https://doi.org/10.1016/j.biopha.2018.12.140 ; http://dx.doi.org/10.1016/j.biopha.2015.12.024

Response 2: The introduction section has been revised accordingly. Details mechanisms involved have been removed as suggested by reviewer 1.

Point 3: Sections: Search strategy and Results are not necessary. Maybe should be removed?

Response 3: We have maintained the search strategy and removed the Prisma flow diagram to eliminate duplicate information. We also maintained a paragraph in Results as a brief description of how the study category was determined. 

Point 4: The Table 1. presents detailed information about essential oils, methods of studies and obtained results. Nevertheless, many of the information are duplicate in Discussion. I recommend improve these two parts. The shortening these parts allow to reduce whole paper what will make it more readable.

Response 4: We have improved the description in the Discussion and Table to reduce the overlap of information.

Point 5: Discussion: The information about MAO inhibition in neurodegeneration should be provide in AD description (see: point 2.)

Response 5: The information on MAO has been re-arranged and included in the Introduction instead of in the Discussion as suggested by the reviewer.

Reviewer 2 Report

The Authors describe a global overview of Essential oils as medicaments for the treatment of neurodegenerative diseases. In particular, this review has been mainly focused on AD and PD according to the great number of articles published in the field.

The manuscript is well organized and discussed. Table 1 is informative. References are adequate.

In my opinion, the information discussed in this manuscript are relevant and will be of interest for the journal readers, but some points should be addressed by the Authors:

- Throughout the manuscript chose to consider EO has a singular or plural name and revise the verbs and pronouns accordingly!

For example, in line 40-41: “EO are volatile 40 and it may play a role in cognitive improvement” should be modified in “EO are volatile 40 and THEY may play a role in cognitive improvement”

- Line 49-52: the sentence is unclear! The ability of EO (but also for all the drugs in general!!!) to improve memory depends on CNS structures organizations, but also to the intrinsic nature of each compound! Please revise its actual form or remove it from text.

- To citations 40 and 41 add also

Villa V. et al Novel celecoxib analogues inhibit glial production of prostaglandinE2, nitric oxide, and oxygen radicals reverting the neuroinflammatoryresponses induced by misfolded prion protein fragment 90-231 or lipopolysaccharide Pharmacological Research 113 (2016) 500–514

- Line 110. Add citations: Nhan, H.S. et al. The multifaceted nature of amyloid precursor protein and its proteolytic fragments: friends and foes. Acta Neuropathol. 2015, 129, 1-19.

Campora, M. et al. Journey on Naphthoquinone and Anthraquinone Derivatives: New Insights in Alzheimer’s Disease. Pharmaceuticals 2021, 14, 33.

- Lines 16, 230 and others: replace the term “evaluate the effectiveness of EO ” with “analyse the effectiveness of EO”

- I would remove the figure 2 since it is properly summarized in the paper main text (line 239-246)

- I'd include a new figure with the most relevant chemical structures and releted anti-neurodegenerative properties identified from EO

- Line 290: replace the term “determinate” with “explore”

- Page 47: please give a more detailed description of the neuroprotective activity of cineole, including the test used in the evaluation

- Page 47: modify the sentence “The most common criteria used in the determination of AD is related to cholinesterases activity” in “The most common criteria used in the determination of AD is related to the anti-cholinesterases activity”

- Near the sentence concluding with REF 188 (page 47), add the main details on active site gorges of both ChEs that are targeted by components of EOs, if in some articles molecular modeling studies have been performed.

- Page 48 Revise the sentence “The slightly longer forms of, particularly Aβ42 which is the lengthier form of Aβ are the principal species deposited in the brain that are more hydro-phobic and fibrillogenic [196]” in “The slightly longer forms of Aβ, particularly Aβ42, are the principal species deposited in the brain that are more hydro-phobic and fibrillogenic [196].”

- Page 49: “The findings indicated that Salvia was the most widely studied EO for AD by using in vitro”. This sentence is uncorrect. Salvia is a genus of plants and not EO; it is a source of essential oils! EO components are studied for the treatment of AD!

- Page 49: “Carvacrol were found to be” modify in “Carvacrol was found to be”

- Page 49 BACE-1, as an example: all abbreviations need to be spelled out in the text!

- In all the text, please use always the same abbreviation Aβ42 or Aβ1-42 also for Aβ40 or Aβ1-40

- Page 58: modify “1,8-cineole effect on neurodegenerative ” with “1,8-cineole effect on neurodegeneration”

Author Response

Point 1: Throughout the manuscript chose to consider EO has a singular or plural name and revise the verbs and pronouns accordingly!

For example, in line 40-41: “EO are volatile 40 and it may play a role in cognitive improvement” should be modified in “EO are volatile 40 and THEY may play a role in cognitive improvement”

Response 1: We have revised the whole manuscript as suggested by the reviewer.

Point 2: Line 49-52: the sentence is unclear! The ability of EO (but also for all the drugs in general!!!) to improve memory depends on CNS structures organizations, but also to the intrinsic nature of each compound! Please revise its actual form or remove it from text.

Response 2: We have removed the text as suggested by the reviewer.

Point 3: - To citations 40 and 41 add also

Villa V. et al Novel celecoxib analogues inhibit glial production of prostaglandinE2, nitric oxide, and oxygen radicals reverting the neuroinflammatoryresponses induced by misfolded prion protein fragment 90-231 or lipopolysaccharide Pharmacological Research 113 (2016) 500–514

Response 3: We have included the citation (Ref no. 35)

Point 4: - Line 110. Add citations: Nhan, H.S. et al. The multifaceted nature of amyloid precursor protein and its proteolytic fragments: friends and foes. Acta Neuropathol. 2015, 129, 1-19.

Campora, M. et al. Journey on Naphthoquinone and Anthraquinone Derivatives: New Insights in Alzheimer’s Disease. Pharmaceuticals 2021, 14, 33.

Response 4: We have included the citation (Ref no 37)

Point 5: - Lines 16, 230 and others: replace the term “evaluate the effectiveness of EO ” with “analyse the effectiveness of EO”

Response 5: We have replaced the term as suggested by the reviewer.

Point 6: - I would remove the figure 2 since it is properly summarized in the paper main text (line 239-246)

Point 6: We have removed Figure 2 as suggested

Point 7: - I'd include a new figure with the most relevant chemical structures and releted anti-neurodegenerative properties identified from EO

Response 7: We have included 10 important chemical structures identified in essential oils (Figure 3).

Point 8:- Line 290: replace the term “determinate” with “explore”

Response 8: We have replaced the term as suggested by the reviewer.

Point 9: - Page 47: please give a more detailed description of the neuroprotective activity of cineole, including the test used in the evaluation

Response 9: We have included the detailed description of the neuroprotective activity of cineole, including the test used in the evaluation in the Discussion under subsection Major components of EOs

Point 10: - Page 47: modify the sentence “The most common criteria used in the determination of AD is related to cholinesterases activity” in “The most common criteria used in the determination of AD is related to the anti-cholinesterases activity”

Response 10:We have replaced the term as suggested by the reviewer.

Point 11: - Near the sentence concluding with REF 188 (page 47), add the main details on active site gorges of both ChEs that are targeted by components of EOs, if in some articles molecular modeling studies have been performed.

Response 11:We have included the term as suggested by the reviewer in the Discussion under subsection Cholinesterase activity in EOs.

Point 12: - Page 48 Revise the sentence “The slightly longer forms of, particularly Aβ42 which is the lengthier form of Aβ are the principal species deposited in the brain that are more hydro-phobic and fibrillogenic [196]” in “The slightly longer forms of Aβ, particularly Aβ42, are the principal species deposited in the brain that are more hydro-phobic and fibrillogenic [196].”

Response 12: We have revised the sentence as suggested by the reviewer.

Point 13: - Page 49: “The findings indicated that Salvia was the most widely studied EO for AD by using in vitro”. This sentence is uncorrect. Salvia is a genus of plants and not EO; it is a source of essential oils! EO components are studied for the treatment of AD!

Response 13: We have revised the sentence accordingly. 

Point 14: - Page 49: “Carvacrol were found to be” modify in “Carvacrol was found to be”

Response 14:We have replaced the term accordingly.

Point 15: - Page 49 BACE-1, as an example: all abbreviations need to be spelled out in the text!

Response 15: We have checked and revised the abbreviation accordingly.

Point 16: - In all the text, please use always the same abbreviation Aβ42 or Aβ1-42 also for Aβ40 or Aβ1-40

Response 16: We have revised the term in the manuscript accordingly.

Point 17: - Page 58: modify “1,8-cineole effect on neurodegenerative ” with “1,8-cineole effect on neurodegeneration”

Response 17: We have revised the term accordingly.

Round 2

Reviewer 1 Report

Authors improved the manuscript in accordance with my suggestions what made the paper more readable. The paper can be accepted for publication. 

Reviewer 2 Report

The Authors have properly improved the manuscript. I consider this manuscript of interest for researchers working in this field, and I suggest its publication in its current form.